# Binary Rating Estimation
# with Graph Side Information

**Kwangjun Ahn**[*]
142nd Military Police Company
Korean Augmentation To the United States Army
kjahnkorea@kaist.ac.kr

**Kangwook Lee**
School of Electrical Engineering
KAIST
kw1jjang@kaist.ac.kr

**Hyunseung Cha**
Kakao Brain
tony.cha@kakaobrain.com

**Changho Suh**
School of Electrical Engineering
KAIST
chsuh@kaist.ac.kr

## Abstract

Rich experimental evidences show that one can better estimate users' unknown ratings with the aid of graph side information such as social graphs. However, the gain is not theoretically quantified. In this work, we study the binary rating estimation problem to understand the fundamental value of graph side information. Considering a simple correlation model between a rating matrix and a graph, we characterize the sharp threshold on the number of observed entries required to recover the rating matrix (called the optimal sample complexity) as a function of the quality of graph side information (to be detailed). To the best of our knowledge, we are the first to reveal how much the graph side information reduces sample complexity. Further, we propose a computationally efficient algorithm that achieves the limit. Our experimental results demonstrate that the algorithm performs well even with real-world graphs.

## 1 Introduction

Recommender systems provide users with appropriate items based on their revealed preference such as ratings and like/dislikes. Due to their wide applicability, recommender systems have received significant attention in machine learning and data mining societies [46, 41, 50, 48, 5, 8, 22, 31]. In addition to the revealed preferences, modern recommender systems also make use of *graph side information* to further improve the performance. For instance, Ma et al. [37] view social networks as user-to-user similarity graph and propose an algorithm that makes use of both revealed ratings and graph side information. As a result, they show that the algorithm can achieve superior performances over those that do not employ social networks. Jamali and Ester [26] demonstrate that an algorithm with graph information can make recommendations for cold start users, whose lack of available rating information precludes the traditional approaches. Similarly, Kalofolias et al. [29] construct an item-to-item similarity graph whose edge weights are computed from the item features, and demonstrate the benefits of exploiting such information.

Apart from the aforementioned, there have been a lot more works that incorporate social graph information in recommender systems; however, few works have been devoted to the theoretical understanding of this problem. In particular, it is widely open as to by how much one can improve the performance with the aid of graph side information. This precisely sets the goal of this paper: we

---

[*]This work was done when Kwangjun Ahn was with KAIST as a student.

aim to quantify the gain due to social network information. Specifically, we intend to characterize the optimal sample complexity needed for rating matrix recovery in the presence of the graph side information.

As an initial effort, we focus on a simple setting in which the entries to be estimated are binary. We consider a scenario where one-sided graph information is available, i.e., either user-to-user or item-to-item graph is given. Without loss of generality, we will assume that a user-to-user similarity graph (or so called a social graph) is available.

Consider $n$ users and $m$ items, where each user rates each item either $+1$ (like) or $-1$ (dislike). The $n$ users are divided into two clusters, and it is assumed that users from the same cluster share their ratings over items. Under this setting, the two types of measurements are available. The first is a partial observation of noisy ratings, and the other is a social graph among the $n$ users, generated as per a celebrated model for random clustered graphs called the stochastic block model (SBM) [21]. Given these, the task is to estimate the ground truth ratings (See Sec. 2 for details). Denote by *the optimal sample complexity* the minimum number of observed ratings required for reliable recovery.

**Main contributions.** The main contributions of this paper are two-fold. First, under the model of interest, we characterize the optimal sample complexity as a function of the quality of graph information; see Sec. 3 for the quantification of the quality. In particular, we quantify by how much the social network information can reduce sample complexity. Our result demonstrates that the social graph can be as significant as resulting in an order-wise reduction in sample complexity.

The second contribution of this work is to develop an efficient algorithm. The algorithm operates in three stages, and the two types of measurements are used separately during the first two stages and then together in the last stage. We provide a theoretical performance guarantee of this algorithm under the model of interest: we prove that the algorithm reliably recovers the ratings as soon as sample complexity exceeds the optimal sample complexity. We also test the empirical performance of the algorithm to show that it achieves better performances than the state-of-the-art approaches [37, 19] even with real-world graphs, including political blog network [4] and Facebook networks [51].

**Related works.** Graph side information has been widely used to improve the performance of recommender systems. To begin with, it has been used in matrix factorization-based (MF) approach, which trains the rating matrix from data by assuming that the rating matrix is of low rank. Most works modify the training procedure by adding some regularization terms inspired by the graph side information [26, 37, 34, 10, 29]. Other than regularization techniques, several works modify the existing rating matrix models using graph information [36, 35, 60, 19]. An online version of this problem is also studied [16]. Another popular approach is the one called neighborhood-based approach, in which user's rating information is predicted based on his/her neighborhoods. Several works [38, 18, 54, 24, 25, 55] improve the performance by properly defining neighborhoods using social graphs. Lastly, some recent works propose deep-learning based approaches in which graph information is incorpated into a framework called graph convolutional network [42, 52].

Recently, few works come up with theoretical guarantees for usuefulness of graph side information. Rao et al. [45] provide a consistency guarantee for regularization techniques, demonstrating a gain due to the side information. On the flip side, Chiang et al. [14] consider a model called dirty inductive matrix completion, which incorporates noisy feature-based side information on top of usual low rank matrix completion model. The theoretical guarantee therein show the efficacy of side information unless it is too noisy. However, whether the gains characterized in both works are the maximally achievable remains open. In this work, focusing on a special case, we are able to characterize optimal sample complexity, the maximum gain due to side information.

Moreover, there have been several recent studies that explore the value of side information in the context of *clustering*. In [39], it is proved that similarity information between data points reduces the adaptive query complexity in exact clustering. A feature-based side information was also considered on top of stochastic block model [47, 59]. A different setting is considered in [6] which demonstrates that the $k$-means problem, which is NP-hard in general, can be solved efficiently with few pairwise queries. In fact, by switching the goal to exactly recover the clusters instead of the ratings, our model can be also seen as a clustering problem with side information. As a consequence of our main result (Theorem 1), we also address this setting; see Corollary 1.

**Notation.** Let $[n] := \{1, 2, \ldots, n\}$; let $\mathbf{1}_{k,\ell}$ be the $k \times \ell$ all-one matrix; for a graph and two subsets $X$ and $Y$, let $e(X, Y)$ be the number of edges between $X$ and $Y$.

## 2 Model

Consider $n$ users and $m$ items, where $m$ can scale with $n$. Each user rates each item either $+1$ (like) or $-1$ (dislike). In real life, people from the same group tend to share their preferences, recommending commodities with each other.[2] In an effort to capture this, we consider a simple setting in which there are two clusters of the same size, and the users from the same cluster share the ratings over items. More precisely, we consider a binary rating matrix $R \in \{+1, -1\}^{n \times m}$ such that half of its rows are $u_R$ and the other half are $v_R$, for some $u_R$ and $v_R$, which we call *rating vectors* of $R$. We denote these two exclusive sets (or clusters) of row indices by $A_R$ and $B_R$, respectively.

**Observation model.** Given $R$, we consider two types of measurements.

*1. Partial observation of noisy ratings ($N^\Omega$)* : We observe each entry of $R$ independently with probability $p$, where $0 \leq p \leq 1$. We further assume that our observed entries are noisy: the value of an observed entry can be flipped with probability $\theta \in (0, \frac{1}{2})$. We denote the subset of observed entries by $\Omega \subset [n] \times [m]$. We represent this with an observation matrix $N^\Omega$ of size $n \times m$, whose $(i, j)$-th entry is the noisy observation if $(i, j) \in \Omega$ and 0 otherwise. In other words, $N_{ij}^\Omega \overset{\text{i.i.d.}}{\sim} R_{ij} \cdot \mathsf{Bern}(p) \cdot (1 - 2\mathsf{Bern}(\theta))$.

*2. Social graph information (G)*: The social graph on $n$ users is generated as per the stochastic block model (SBM) [21]. That is, given the two clusters of users $A_R$ and $B_R$, an edge between each pair of two users $i, j$ is placed with probability $\alpha$, independently of the others, if they are from the same cluster. If not, the probability of having an edge between them is $\beta$, where $\alpha \geq \beta$. Let $G = ([n], E)$ denote the social graph.

**Performance metric.** Given $N^\Omega$ and $G$, the task of interest is to recover $R$. The performance of an estimator is measured by the probability that the output of estimator does not coincide with $R$, namely *the probability of error*. Concretely, we assume that the worst-case matrix is chosen from a collection of rating matrices $R'$ with $\|u_{R'} - v_{R'}\|_0 = \lceil \gamma m \rceil$, where $\gamma \in (0, 1)$ is a fixed constant.

**Definition 1.** *For a fixed constant $\gamma \in (0, 1)$ and an estimator $\psi$ that outputs a binary rating matrix based on $N^\Omega$ and $G$, the worst-case probability of error $P_e^{(\gamma)}(\psi)$ is defined as $P_e^{(\gamma)}(\psi) := \max_{R' : \|u_{R'} - v_{R'}\|_0 = \lceil \gamma m \rceil} \Pr\left(\psi(N^\Omega, G) \neq R'\right).$*

**Goal.** We aim to characterize $p^\star$ such that (i) when $p$ exceeds $p^\star$, $P_e^{(\gamma)}(\psi) \to 0$ as $n \to \infty$ for some estimator $\psi$; (ii) when $p$ is less than $p^\star$, $P_e^{(\gamma)}(\psi) \not\to 0$ for any $\psi$. In particular, we aim to characterize $nmp^\star$ that we call *the optimal sample complexity*. Given the fact that $nmp$ is the expected number of observed entries, the optimal sample complexity can be seen as the minimum number of observed entries required for rating recovery in the limit of $n$.

## 3 Optimal sample complexity

We characterize the optimal sample complexity as a function of $n, m, \theta, \gamma, \alpha$, and $\beta$.

**Theorem 1.** *Let $\gamma \in (0, 1)$ be fixed. Assume that $m = \omega(\log n)$ and $\log m = o(n)$.[3] Then, the following holds for any constant $\epsilon > 0$: if*

$$p \geq \frac{1}{(\sqrt{1-\theta} - \sqrt{\theta})^2} \max\left\{ \frac{(1+\epsilon)\log n - \frac{n}{2}(\sqrt{\alpha} - \sqrt{\beta})^2}{\gamma m}, \frac{(1+\epsilon)2\log m}{n} \right\},$$

*then $P_e^{(\gamma)}(\psi) \to 0$ as $n \to \infty$ for some $\psi$ that outputs a binary rating matrix based on $N^\Omega$ and $G$ ; conversely, if*

$$p \leq \frac{1}{(\sqrt{1-\theta} - \sqrt{\theta})^2} \max\left\{ \frac{(1-\epsilon)\log n - \frac{n}{2}(\sqrt{\alpha} - \sqrt{\beta})^2}{\gamma m}, \frac{(1-\epsilon)2\log m}{n} \right\},$$

*then $P_e^{(\gamma)}(\psi) \not\to 0$ as $n \to \infty$ for any $\psi$.*

*Proof:* See Sec. 5 for the proof sketch; and see the supplemental material for the full proof .  □

Let us interpret Theorem 1. See Table 1 for a summary. In essence, Theorem 1 asserts that the rating recovery is possible if and only if

$$p > p^\star := \frac{1}{(\sqrt{1-\theta} - \sqrt{\theta})^2} \max\left\{(\gamma m)^{-1}\log n - (2\gamma m)^{-1}n(\sqrt{\alpha} - \sqrt{\beta})^2, \; n^{-1}2\log m\right\}.$$

For illustrative purpose, we introduce a notation: $I_s := (\sqrt{\alpha} - \sqrt{\beta})^2$. One can interpret $I_s$ as the quality of social graph information. This is because the two-cluster structure becomes more transparent as the gap between $\alpha$ and $\beta$ gets larger. For instance, when $\alpha = \beta$, i.e., $I_s = 0$, there is no way to distinguish the two clusters. On the other hand, when $\alpha = 1$ and $\beta = 0$ (or $\alpha = 0$ and $\beta = 1$), i.e., $I_s = 1$, the cluster structure is straightforward from $G$. Note that the notation $I_s$ is also employed in the context of community recovery under SBM, in which the fundamental limit for exact recovery is shown to be $I_s > 2\frac{\log n}{n}$ [1].

First, consider $I_s = 0$. In this case, the optimal sample complexity $nmp^\star$ is

$$\frac{1}{(\sqrt{1-\theta} - \sqrt{\theta})^2} \max\{\gamma^{-1}n\log n, 2m\log m\}. \tag{1}$$

**Remark 1.** *Note that the fundamental limit decreases in $\gamma$. This tendency is intuitive. To see this, focus on the noiseless setting ($\theta = 0$). Let us classify entries of the rating matrix into two types: (i) the entries on the columns where the ratings of two groups coincide; and (ii) the other entries. Unlike the first type of entries, the second type of entries have possibility to reveal significant information on clusters: when we observe two users' different ratings on the same item, it can be immediately concluded that the two users belong to different clusters. Hence, the second type of entries are more informative. As there are $\gamma$-fraction of entries of the second type, the chance of getting more informative entries increases in $\gamma$. Thus, the required sample complexity decreases in $\gamma$.*

We now turn to the case $I_s \neq 0$. In this case, the optimal sample complexity $nmp^\star$ is

$$\frac{1}{(\sqrt{1-\theta} - \sqrt{\theta})^2} \max\{\gamma^{-1}n\log n - (2\gamma)^{-1}n^2 I_s, \; 2m\log m\}. \tag{2}$$

On the one hand, this result suggests that the social graph information does not help rating recovery when the number of users is relatively smaller than that of items. More precisely, when $2n\log n \leq 4\gamma m\log m$, both (1) and (2) are equal to $2m\log m/(\sqrt{1-\theta} - \sqrt{\theta})^2$.

On the other hand, when $2n\log n > 4\gamma m\log m$, i.e., (1) is equal to $\gamma^{-1}n\log n/(\sqrt{1-\theta} - \sqrt{\theta})^2$, the result implies that the social graph information does help rating recovery. Below, we examine the amount of reduction in sample complexity as a function of $I_s$. For simplicity, we focus on a setting in which $\theta = 0$ and $\gamma = \frac{1}{2}$, i.e., (1) is equal to $2n\log n$.

We first consider the case of $n^2 I_s < 2n\log n - 2m\log m$, i.e., (2) being equal to $2n\log n - n^2 I_s$. In this case, sample complexity is reduced by $n^2 I_s$. That being said, there is no asymptotical gain unless $n^2 I_s = \Omega(n\log n)$. On the other hand, when $n^2 I_s = \Omega(n\log n)$, this reduction can be as significant as resulting in an order-wise reduction in sample complexity. To see this clearly, consider two scenarios: $n = 2m$ and $n = m^2$. Also see Fig. 1.

**Example 1** ($n = 2m$). *Note that $n^2 I_s < 2n\log n - 2m\log m$ if and only if $I_s < \log(2n)/n$. When $I_s = c\log(2n)/n$ for $0 < c < 1$, the optimal sample complexity is $2n\log n - cn\log 2n = (2 - c)n\log n - cn\log 2$, which is (asymptotically) lower than (1) by a multiplicative factor of $\frac{c}{2}$.*

**Example 2** ($n = m^2$). *Note that $n^2 I_s < 2n\log n - 2m\log m$ if and only if $I_s < 2\log n/n - \log n/n^{1.5}$. When $I_s = 2\log n/n - \log n/n^c$ for $1 < c < 1.5$, the optimal sample complexity is $n^{2-c}\log n$, which shows an order-wise reduction in sample complexity.*

**Remark 2.** *Example 2 justifies the observation made by Jamali and Ester [26] that graph side information can help predict ratings for cold start users. More specifically, note that when $c = 1.4$, most users are cold start users as the average number of observed ratings per user is $n^{-0.4}\log n$.*

For the case of $n^2 I_s \geq 2n\log n - 2m\log m$, (2) $= 2m\log m$ no matter how large $I_s$ is. This implies that the gain due to side information is saturated.

Table 1: **Summary of the gain in sample complexity.** Depending on the quality of graph information $I_s := (\sqrt{\alpha} - \sqrt{\beta})^2$, the gain in sample complexity can be summarized as follows. First, graph information does not help unless the number of users is relatively larger than that of items ($2n \log n \geq 4\gamma m \log m$). When it helps, the efficacy of the information depends on its quality: When $n^2 I_s = o(n \log n)$, social network is too noisy, and hence does not help rating recovery. When $I_s = \Omega(\log n/n)$, the minimum sample complexity is a decreasing function in $I_s$. In other words, as the quality of social network increases, the minimum sample complexity decreases. However, the gain of side-information is characterized as diminishing returns: When $I_s$ is larger than a certain threshold, the minimum sample complexity stops decreasing. Note that this does not mean that graph information does not help: It helps but its gain is saturated.

| Value of $n^2 I_s$ | | |
|---|---|---|
| $o(n \log n)$ | $< 2n \log n - 4\gamma m \log m$ | $\geq 2n \log n - 4\gamma m \log m$ |
| no asymptotical gain | gain increases in $I_s$ | gain is saturated. |

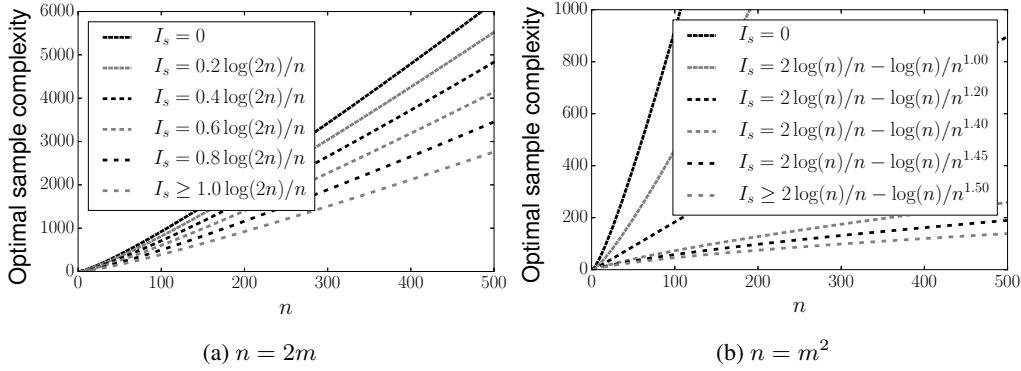

(a) $n = 2m$                                       (b) $n = m^2$

Figure 1: **Significant reduction in sample complexity due to social graph.** We illustrate Theorem 1 for two cases: $n = 2m$ and $n = m^2$. For $n = 2m$, the optimal sample complexity is reduced by some multiplicative factors, and for $n = m^2$, there is an order-wise reduction in sample complexity. See Example 1 and 2 for details.

**Implications on community recovery.** Switching the goal of our model to exactly recover $A_R$ and $B_R$ instead of $R$, our model can cover a community recovery problem with some side information with respect to ratings. The proof of Theorem 1 suggests the fundamental limit on $I_s$ for exact recovery:

**Corollary 1.** *Suppose we wish to exactly recover the clusters $A_R$ and $B_R$ (up to a flip) instead of $R$ (we should also modify the definition of $P_e^{(\gamma)}(\psi)$ accordingly). Then, the following holds for any constant $\epsilon > 0$: $P_e^{(\gamma)}(\psi) \to 0$ as $n \to \infty$ for some $\psi$ that outputs two equal-sized clusters based on $N^{\Omega}$ and $G$ whenever $I_s + \frac{2\gamma m p(\sqrt{1-\theta} - \sqrt{\theta})^2}{n} \geq (1+\epsilon)\frac{2\log n}{n}$; conversely, $P_e^{(\gamma)}(\psi) \not\to 0$ as $n \to \infty$ for any $\psi$ if $I_s + \frac{2\gamma m p(\sqrt{1-\theta} - \sqrt{\theta})^2}{n} \leq (1-\epsilon)\frac{2\log n}{n}$.*

This result implies that when $p \neq 0$, the amount of graph information $I_s$ required for cluster recovery reduces from $\frac{2\log n}{n}$ [1] to $\max\left\{ \frac{2\log n}{n} - \frac{2\gamma m p(\sqrt{1-\theta} - \sqrt{\theta})^2}{n}, 0 \right\}$.

## 4 Proposed algorithm

In this section, we propose an efficient rating estimation algorithm, which achieves the fundamental limit characterized in Theorem 1. We note that our algorithm can also be applied to a general setting (not limited to the simple two-cluster setting described earlier), although the theoretical guarantee for the setting is not provided. This will be clearer while describing our algorithm.

**Algorithm description.** The algorithm works in three stages. The inputs of the algorithm are $N^{\Omega}$, $G$, the number of clusters $k$ and hyperparameters $c_1, c_2 > 0$.[4]

*Stage 1 (Partial recovery of clusters)*: First, we run a spectral method [17] on $G$. Let $A_1^{(0)}, A_2^{(0)}, \ldots, A_k^{(0)}$ be the output of the clustering. Note that other clustering algorithms such as other spectral methods [2, 15, 33], nonbacktracking matrix based methods [32], semidefinite programming (SDP) [27], and belief propagation (BP) variants [43] can be employed for this stage.

*Stage 2 (Recovery of rating vectors)*: Next, for each $j$, we recover the rating vector of the cluster $A_j^{(0)}$ using the observed ratings. For each item $i$, we collect the observed ratings of the item by the users in $A_j^{(0)}$. Among the collected ratings, we find the one that appears most frequently; let the rating be $u_i^{(j)}$. The rating vector is then defined as $u^{(j)} = \left[ u_i^{(j)} \right]_{i=1}^m$.

*Stage 3 (Local refinement of clusters)*: The last stage iteratively refines the clusters $A_j^{(0)}$'s using $G$, $N^\Omega$ and $u^{(j)}$'s. This stage consists of $T$ times of refinement steps, and in each step, the clusters are updated as follows. Let $A_1^{(t-1)}, A_2^{(t-1)}, \ldots, A_k^{(t-1)}$ be the outcome of $(t-1)$-th refinement step for some $t = 1, 2, \ldots, T$. Then, for each $i = 1, 2, \ldots, n$, we put user $i$ to $A_{j^*}^{(t)}$, which is the one among $A_j^{(t-1)}$'s that gives the maximum value of

$$c_1 \cdot e\left( \{i\}, A_j^{(t-1)} \right) - c_2 \cdot \bar{e}\left( \{i\}, A_j^{(t-1)} \right) + \Pi_i(u^{(j)}).$$

Here, $\bar{e}(\{i\}, A_j^{(t-1)}) := |A_j^{(t-1)}| - e(\{i\}, A_j^{(t-1)})$ indicates the number of non-edges between $i$ and $A_j^{(t-1)}$; $\Pi_i(u^{(j)})$ is the number of user $i$'s observed ratings which coincide with that of $u^{(j)}$.

Lastly, the algorithm outputs $\widehat{R}$ where $i$-th row is $u^{(j)}$ whenever $i \in A_j^{(T)}$.

**Remark 3** (Update rule in Stage 3). *For a cluster $A_j^{(t-1)}$ and its rating vector $u^{(j)}$, the term $c_1 e\left( \{i\}, A_j^{(t-1)} \right) + \Pi_i(u^{(j)})$, which is the sum of the first and the third terms of the update rule, can be seen as a measure of* fitness *of user $i$ with respect to the cluster $A_j^{(t-1)}$. This is because $i$ is more likely to belong to $A_j^{(t-1)}$ when there are more edges between $i$ and $A_j^{(t-1)}$ and more observed ratings consistent with $u^{(j)}$. The number of non-edges is subtracted from the term to minimize the effect of cluster size as a large cluster tends to have more edges to users.*

**Remark 4** (Non-agnostic refinement rule). *When the algorithm knows the size of each cluster, we modify the definition of $\bar{e}$ by replacing the terms $|A_j^{(t-1)}|$'s with the actual sizes $|A_j|$'s. In particular, when the algorithm knows that the clusters are of equal size, we obtain the following refinement rule: we put user $i$ to $A_{j^*}^{(t)}$, where $j^* = \arg\max_{j \in [k]} \left[ c_1' \cdot e\left( \{i\}, A_j^{(t-1)} \right) + \Pi_i(u^{(j)}) \right]$ for some hyperparameter $c_1' > 0$.*

**Remark 5.** *Note that this algorithm can be applied to a general setting where (i) rating type is not limited to binary; (ii) the number of clusters can be larger than 2; and (iii) the clusters are of unequal sizes.*

**Remark 6.** *The proposed algorithm is inspired by a general paradigm in solving non-convex problems: first obtain a decent initial estimate and iteratively refine the estimate to reach the global optimum. This paradigm has been employed in various contexts, including matrix completion [30, 23], phase retrieval [44, 11], robust PCA [56], community recovery [2, 17, 57, 12], EM-algorithm [7], and rank aggregation [13]. Moreover, we note that spectral algorithms have been also used in rating estimation in the context of crowdsourcing [49, 53].*

One important aspect of this algorithm is its low computational complexity. Spectral methods can be run within $O(|E| \log n)$ time using the power method [9]. Stage 2 requires a single pass of all observed ratings, which amounts to $O(|\Omega|)$ time. As for Stage 3, a single individual update of user $i$ entails reading of $i$-th row and edges connected to user $i$. Assuming a proper data structure, each iteration requires $O(|E| + |\Omega|)$. Hence, Stage 3 can be done within $O(|E|T + |\Omega|T)$. Overall, the proposed algorithm can be performed within $O(|E|T + |E| \log n + |\Omega|T)$.

**Theoretical performance guarantee.** To investigate theoretical guarantees of the proposed algorithm, we focus on the model in Sec. 2.

**Theorem 2.** *Let $R$ be any binary rating matrix with $\|u_R - v_R\|_0 = \gamma m$ for some $\gamma \in (0,1)$. In addition to the assumptions in Theorem 1, assume further that $m = O(n)$ and $I_s = \omega(1/n)$.[5] Suppose that there exists $\epsilon > 0$ such that the sufficient condition in Theorem 1 holds. Then with probability approaching $1$ as $n \to \infty$, the proposed algorithm with the knowledge that the two clusters are of equal size (i.e., using the refinement rule in Remark 4) exactly recovers $R$ under the settings $T = O(\log n)$ and $c_1' = \log\left(\frac{\hat{\alpha}(1-\hat{\beta})}{\hat{\beta}(1-\hat{\alpha})}\right) / \log\left(\frac{1-\hat{\theta}}{\hat{\theta}}\right)$. Here, $\hat{\alpha} := \frac{e(A_1(0),A_1(0))+e(A_2(0),A_2(0))}{2\binom{n/2}{2}}$ and $\hat{\beta} := 4\frac{e(A_1^{(0)},A_2^{(0)})}{n^2}$ are estimations of model parameters $\alpha$ and $\beta$ after Stage 1, and $\hat{\theta}$ is an estimation of $\theta$ after Stage 2 defined by the fraction of observed ratings which are different from the corresponding entries of the rating matrix defined by clusters $A_1^{(0)}, A_2^{(0)}$ and rating vectors $u^{(1)}, u^{(2)}$.*

*Proof:* Due to space limitation, we defer the proof to the supplemental material. □

Theorem 2 implies that the proposed algorithm with proper hyperparameter choices achieves the fundamental limit in Theorem 1 except for the case of scarce social graph information ($I_s = O(1/n)$).

## 5   Proof outline of Theorem 1

We sketch the proof while deferring the full proof to the supplemental material. Let $I_r := p(\sqrt{1-\theta} - \sqrt{\theta})^2$. Using the notations of $I_r$ and $I_s$, one can succinctly represent the sufficient condition and the necessary condition claimed in Theorem 1. For instance, the sufficient condition reads

$$\frac{1}{2}nI_s + \gamma m I_r \geq (1+\epsilon)\log n \qquad \text{and} \qquad \frac{1}{2}nI_r \geq (1+\epsilon)\log m \,.$$

We next introduce a few more notations that will be used throughout the proof. Let $\mathcal{C}^{(\gamma)}$ be the collection of rating matrices $R$ such that $\|u_R - v_R\|_0 = \gamma m$ (Here and below, we treat $\gamma m$ as an integer for notational simplicity); let $R^{(\gamma)} := \left[\begin{array}{c|c} +\mathbf{1}_{n/2,(1-\gamma)m} & +\mathbf{1}_{n/2,\gamma m} \\ \hline +\mathbf{1}_{n/2,(1-\gamma)m} & -\mathbf{1}_{n/2,\gamma m} \end{array}\right] \in \mathcal{C}^{(\gamma)}$ (i.e., $A_{R^{(\gamma)}} = [\frac{n}{2}], B_{R^{(\gamma)}} = [n] \setminus [\frac{n}{2}], u_{R^{(\gamma)}} = +\mathbf{1}_{1,m}$ and $v_{R^{(\gamma)}} = [+\mathbf{1}_{1,(1-\gamma)m} \mid -\mathbf{1}_{1,\gamma m}]$). Lastly, let $\psi_{\mathrm{ML}}$ be the maximum likelihood estimator (output being not constrained to $\mathcal{C}^{(\gamma)}$) and $\mathsf{L}(\cdot)$ be the likelihood function.

**Achievability:** It is enough to show that $P_e^{(\gamma)}(\psi_{\mathrm{ML}}) \to 0$. By symmetry, we fix the ground truth rating matrix to be $R^{(\gamma)}$. Note that the event "$\psi_{\mathrm{ML}}(N^\Omega, G) \neq R^{(\gamma)}$" happens only if $\mathsf{L}(X) \leq \mathsf{L}(R^{(\gamma)})$ for some binary rating matrix $X$. Hence, by the union bound,

$$P_e^{(\gamma)}(\psi_{\mathrm{ML}}) \leq \sum_{X \neq M} \Pr\left(\mathsf{L}(X) \leq \mathsf{L}(R^{(\gamma)})\right) \,. \tag{3}$$

To enumerate all rating matrices different from $R^{(\gamma)}$, we define $\mathcal{X}(k, a_1, a_2, b_1, b_2)$ to be the class of rating matrices $X$'s such that (i) $|A_X \setminus A_{R^{(\gamma)}}| = |B_X \setminus B_{R^{(\gamma)}}| =: k$; (ii) $u_X$ differs from $u_{R^{(\gamma)}}$ at $a_1$ many coordinates among the first $(1-\gamma)m$ coordinates and at $a_2$ many coordinates among the next $\gamma m$ coordinates; and (iii) $v_X$ differs from $v_{R^{(\gamma)}}$ at $b_1$ many coordinates among the first $(1-\gamma)m$ coordinates and at $b_2$ many coordinates among the next $\gamma m$ coordinates. Note that if $X_1$ and $X_2$ belong to the same class, then

$$\Pr\left(\mathsf{L}(X_1) \leq \mathsf{L}(M)\right) = \Pr\left(\mathsf{L}(X_2) \leq \mathsf{L}(M)\right)$$

as the two events are statistically identical. Let $\mathcal{I}$ be the range of index, i.e., collection of tuples $(k, a_1, a_2, b_1, b_2) \neq (0,0,0,0,0)$ such that $0 \leq k \leq n/4$ and $0 \leq a_1, b_1 \leq (1-\gamma)m$ and $0 \leq a_2, b_2 \leq \gamma m$. Note that $k \leq n/4$ is sufficient as one can switch the role of $u_X$ and $v_X$.

For each 5-tuple $z \in \mathcal{I}$, let $X_z$ be a binary rating matrix in $\mathcal{X}(z)$. With this enumeration, the right hand side of (3) becomes $\sum_{z \in \mathcal{I}} |X(z)| \Pr\left(\mathsf{L}(X_z) \leq \mathsf{L}(M)\right)$.

Let $z = (k, a_1, a_2, b_1, b_2)$. To upper bound $\Pr\left(\mathsf{L}(X_z) \leq \mathsf{L}(M)\right)$, we developed a large deviation result building upon the techniques in [28, 58]: for $z = (k, a_1, a_2, b_1, b_2)$,

$$\Pr\left(\mathsf{L}(X_z) \leq \mathsf{L}(M)\right) \leq e^{-2(\frac{n}{2}-k)kI_s - \mathcal{D}_z I_r} \,, \tag{4}$$

where $\mathcal{D}_z := k \cdot \{a_1 + b_1 + (\gamma m - a_2) + (\gamma m - b_2)\} + \left(\frac{n}{2} - k\right) \cdot (a_1 + a_2 + b_1 + b_2)$; we refer readers to the supplemental material for details. Let $S(z) := |X(z)|e^{-2(\frac{n}{2} - k)kI_s - \mathcal{D}_z I_r}$. Then, the last upper bound is bounded by $\sum_{z \in \mathcal{I}} S(z)$.

In the supplemental material, we show that $S(z)$'s for $z$ with at least one large coordinate are negligible. More precisely, for $\mathcal{L} := \{(k, a_1, a_2, b_1, b_2) : k < \delta n \text{ and } a_1, a_2, b_1, b_2 < \delta m\}$ (where $\delta$ is a sufficiently small quantity), $\sum_{\mathcal{I} \setminus \mathcal{L}} S(z)$ is negligible compared to $\sum_{\mathcal{I} \cap \mathcal{L}} S(z)$. The rationale behind this is that when $z \in \mathcal{I} \setminus \mathcal{L}$, $S(z)$ becomes a very small quantity as either $2(\frac{n}{2} - k)k$ or $\mathcal{D}_z$ becomes very large.

Hence, it suffices to focus on $\sum_{\mathcal{I} \cap \mathcal{L}} S(z)$. As $k \ll n$ and $a_1, a_2, b_1, b_2 \ll m$ when $z \in \mathcal{I} \cap \mathcal{L}$, one can approximate $2(\frac{n}{2} - k)k \approx nk$ and $\mathcal{D}_z \approx 2\gamma km + \frac{n}{2}(a_1 + a_2 + b_1 + b_2)$. By definition,

$$|\mathcal{X}(k, a_1, a_2, b_1, b_2)| = \binom{\frac{n}{2}}{k}^2 \binom{(1-\gamma)m}{a_1} \binom{(1-\gamma)m}{b_1} \binom{\gamma m}{a_2} \binom{\gamma m}{b_2} \le n^{2k} m^{a_1 + a_2 + b_1 + b_2}.$$

This together with the above approximation yields

$$S(z) \le e^{2k \log n + (a_1 + a_2 + b_1 + b_2) \log m} e^{-2(\frac{n}{2} - k)kI_s - \mathcal{D}_z I_r}$$

$$\approx e^{-k \cdot (nI_r + 2\gamma mI_s - 2\log n) - (a_1 + a_2 + b_1 + b_2) \cdot (\frac{n}{2}I_r - \log m)} \le (n^{-2\epsilon})^k (m^{-\epsilon})^{(a_1 + a_2 + b_1 + b_2)},$$

where the last inequality is due to $\frac{1}{2}nI_s + \gamma mI_r \ge (1 + \epsilon)\log n$ and $\frac{1}{2}nI_r \ge (1 + \epsilon)\log m$. This justifies $\sum_{z \in \mathcal{I} \cap \mathcal{L}} S(z) \to 0$.

**Converse:** *Step 1 (ML as an optimal estimator):* Consider the maximum likelihood estimator $\psi_{\mathrm{ML}}|_{\mathcal{C}^{(\gamma)}}$ whose output is constrained in $\mathcal{C}^{(\gamma)}$. It can be proven that

$$\inf_\psi P_e^{(\gamma)}(\psi) \ge P_e^{(\gamma)}(\psi_{\mathrm{ML}}|_{\mathcal{C}^{(\gamma)}}).$$

See the supplemental material for details. Hence, it is enough to show $P_e^{(\gamma)}(\psi_{\mathrm{ML}}|_{\mathcal{C}^{(\gamma)}}) \nrightarrow 0$. By symmetry, we fix the ground truth rating matrix to be $R^{(\gamma)}$.

*Step 2 (Genie-aided ML estimators):* We consider *genie-aided* ML estimators, in which the genie helps the estimator by telling the answer is one of few candidates within $\mathcal{C}^{(\gamma)}$. Owing to the notation $\mathcal{X}(\cdot, \cdot, \cdot, \cdot, \cdot)$ in the achievability proof, two different kinds of genie-aided estimators are examined: $\psi_{\mathrm{ML}}^{(1)}$ is given with the information that the ground truth belongs to $R^{(\gamma)} \cup \mathcal{X}(0, 1, 1, 0, 0)$; $\psi_{\mathrm{ML}}^{(2)}$ is given with the information that the ground truth belongs to $R^{(\gamma)} \cup \mathcal{X}(1, 0, 0, 0, 0)$.

*Step 3 (Analysis of genie-aided estimators):* We prove (i) $\psi_{\mathrm{ML}}^{(1)}$ fails if $\frac{1}{2}nI_r \le (1 - \epsilon)\log m$ and (ii) $\psi_{\mathrm{ML}}^{(2)}$ fails if $\frac{1}{2}nI_s + \gamma mI_r \le (1 - \epsilon)\log n$. Here, we provide the proof sketch of (i): we remark that the proof of (ii) is trickier and it requires some combinatorial properties of random graphs. Note that if the likelihood $\mathsf{L}(X)$ for some $X \in \mathcal{X}(0, 0, 0, 1, 1)$ is less than or equal to $\mathsf{L}(R^{(\gamma)})$, then $\psi_{\mathrm{ML}}^{(1)}$ fails with probability at least $1/2$. Hence, it is enough to show that with probability approaching 1, there exists $X \in \mathcal{X}(0, 0, 0, 1, 1)$ such that $\mathsf{L}(X) \le \mathsf{L}(R^{(\gamma)})$, or equivalently (by taking complement), $\Pr\left(\bigcap_{X \in \mathcal{X}(0,0,0,1,1)} [\mathsf{L}(X) > \mathsf{L}(R^{(\gamma)})]\right) \to 0$. On the other hand, some difficulties arise while analysing the last probability as the events $[\mathsf{L}(X) > \mathsf{L}(R^{(\gamma)})]_{X \in \mathcal{X}(0,0,0,1,1)}$ are not mutually independent. A trick avoiding this issue is to show that the last probability is bounded by $\Pr\left(\bigcap_{X \in \mathcal{X}(0,0,0,1,0)} [\mathsf{L}(X) > \mathsf{L}(R^{(\gamma)})]\right) + \Pr\left(\bigcap_{X \in \mathcal{X}(0,0,0,0,1)} [\mathsf{L}(X) > \mathsf{L}(R^{(\gamma)})]\right)$. The analysis is now tractable as the collections of events $[\mathsf{L}(X) > \mathsf{L}(R^{(\gamma)})]_{X \in \mathcal{X}(0,0,0,1,0)}$ and $[\mathsf{L}(X) > \mathsf{L}(R^{(\gamma)})]_{X \in \mathcal{X}(0,0,0,0,1)}$ are both mutually independent. Now, we conclude the proof by using the reverse direction of the bound (4) (the bound (4) is indeed tight and the reverse direction also holds up to a constant factor; see the supplemental material): For instance,

$$\Pr\left(\bigcap_{X \in \mathcal{X}(0,0,0,1,0)} [\mathsf{L}(X) > \mathsf{L}(R^{(\gamma)})]\right) \le (1 - e^{-\frac{n}{2}I_r})^{|\mathcal{X}(0,0,0,1,0)|} \le e^{-(1-\gamma)me^{-\frac{n}{2}I_r}},$$

where the last inequality is due to the inequality $1 - x \le e^{-x}$ and $|\mathcal{X}(0, 0, 0, 1, 0)| = (1 - \gamma)m$; the last term goes to zero when $\frac{1}{2}nI_r \le (1 - \epsilon)\log m$.

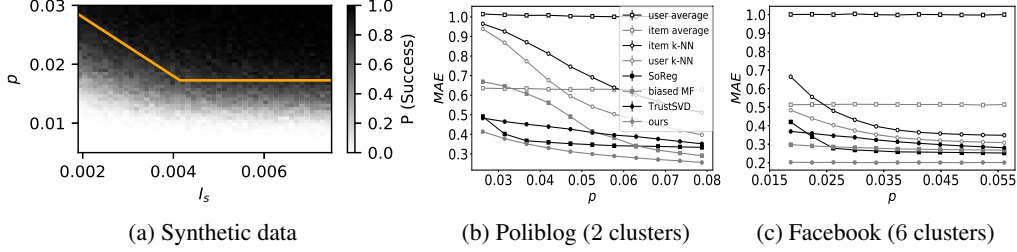

|  | (a) Synthetic data | (b) Poliblog (2 clusters) | (c) Facebook (6 clusters) |

Figure 2: (a) The level of darkness depicts the empirical success rate, and the orange line reflects the optimal sample complexity due to Theorem 1. A sharp transition in darkness near the line corroborates Theorem 1. (b&c) Performance comparison of algorithms on real datasets. Our algorithm shows better performance than the other algorithms on every data set, demonstrating practicality of our approach.

## 6 Experiments

We first conduct an experiment to corroborate Theorem 1. We consider a setting where $n = 2000$ users and $m = 1000$ items. The synthetic data is generated as per the model in Sec. 2.[6] For each triple $(p, \alpha, \beta)$, an empirical success rate of the proposed algorithm is measured over 100 random trials. Figure 2a shows the result, where the empirical success rate is depicted by the level of darkness. The orange (solid) line reflects the optimal sample complexity due to Theorem 1. The phase transition occurs near the orange line, corroborating Theorem 1.

We next show that our proposed algorithm performs well even with real-world graphs. On top of the real graphs (political blog network [4] and Facebook networks [51]), we synthesize ratings as per our model. For the performance metric, we use mean absolute error (MAE): $\mathbb{E}[|\widehat{R}_{ij} - R_{ij}|]$, where the expectation is over the observed ratings. We then compare the performance of our algorithm with 7 well-known recommendation algorithms. Specifically, we compare the performance of our algorithm with 7 recommendation algorithms.[7] Reported in Figure 2b, 2c are the performances of rating estimation algorithms on real graph data. Our algorithm shows better performance than the other algorithms, showing the practicality of our approach.

## 7 Conclusion

Motivated by the lack of study in quantifying the value of social graph information in recommender systems, this work characterized the optimal sample complexity of the rating recovery problem with social graph information. We also proposed an efficient rating estimation algorithm that provably achieves the optimal sample complexity.

This paper comes with some limitations in characterizing sample complexity for more general models. We hope restrictive assumptions considered in this paper, such as binary ratings and rating being shared across the same group, be relaxed in the future endeavors. In particular, it would be interesting to characterize the optimal sample for feature-based side information models [14, 45]. Moreover, as in the case of community detection, sample complexity for partial recovery [3] might be more desirable in practice over our exact recovery setting.

### Acknowledgments

The work was jointly supported by the National Research Foundation of Korea (NRF) grant funded by the Korea government (MSIP) (No. 2018R1A1A1A05022889) and Kakao Brain Corp.

## Footnotes

[2]This tendency, called *homophily*, has been extensively studied in sociology and psychology [40].

[3]We employ these conditions to obtain the large deviation results in the proof (See the supplemental material). Intuitively, these conditions rule out tall and fat matrices, respectively.

[4] One can always use validation data to tune $c_1$ and $c_2$. For the case of two equal-sized communities, we characterized the optimal choice of $c_1$ and $c_2$ in Theorem 2.

[5]Note that the condition $m = O(n)$ is for reliable estimation of parameters $\alpha$, $\beta$, $\theta$, and hence can be removed when the parameters are known.

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
