[Supplementary Material]



# Binary Rating Estimation with Graph Side Information: Supplementary Material

## I. PROOF OF OPTIMAL SAMPLE COMPLEXITY (THEOREM 1)

We first introduce a few more notations that will be used throughout the proof. Let $\mathcal{C}^{(\gamma)}$ be the collection of rating matrices $R$ such that $\|u_R - v_R\|_0 = \gamma m$. Let

$$R^{(\gamma)} := \left[ \begin{array}{c|c} +\mathbf{1}_{n/2,(1-\gamma)m} & +\mathbf{1}_{n/2,\gamma m} \\ \hline +\mathbf{1}_{n/2,(1-\gamma)m} & -\mathbf{1}_{n/2,\gamma m} \end{array} \right] \in \mathcal{C}^{(\gamma)} \tag{1}$$

(i.e., $A_{R^{(\gamma)}} = [\frac{n}{2}]$, $B_{R^{(\gamma)}} = [n] \setminus [\frac{n}{2}]$, $u_{R^{(\gamma)}} = +\mathbf{1}_{1,m}$ and $v_{R^{(\gamma)}} = [+\mathbf{1}_{1,(1-\gamma)m} \mid -\mathbf{1}_{1,\gamma m}]$). Lastly, let $I_r := p(\sqrt{1-\theta} - \sqrt{\theta})^2$. Using the notations of $I_r$ and $I_s$, one can succinctly write the conditions in Theorem 1:

- The sufficient condition $\Leftrightarrow \frac{1}{2} n I_s + \gamma m I_r \geq (1+\epsilon) \log n$ and $\frac{1}{2} n I_r \geq (1+\epsilon) \log m$;
- The necessary condition $\Leftrightarrow \frac{1}{2} n I_s + \gamma m I_r \leq (1-\epsilon) \log n$ or $\frac{1}{2} n I_r \leq (1-\epsilon) \log m$.

Since the probability of error decreases as $p$, it suffices to prove for the boundary case $p = O\left( \frac{\log n}{m} + \frac{\log m}{n} \right)$. Similarly, we consider $I_s = O\left( \frac{\log n}{n} \right)$, i.e., $\alpha, \beta = O\left( \frac{\log n}{n} \right)$.

### A. MLE Achievability

We will show that the maximum likelihood estimator $\psi_{\mathrm{ML}}$ satisfies $P_e^{(\gamma)}(\psi_{\mathrm{ML}}) \to 0$ if $\frac{1}{2} n I_s + \gamma m I_r \geq (1+\epsilon) \log n$ and $\frac{1}{2} n I_r \geq (1+\epsilon) \log m$. As the event "$\psi_{\mathrm{ML}}(N^\Omega, G) \neq R \mid R = X$" is statistically identical for all $X \in \mathcal{C}^{(\gamma)}$, we have

$$P_e^{(\gamma)}(\psi_{\mathrm{ML}}) = \max_{R \in \mathcal{C}^{(\gamma)}} \Pr\left( \psi_{\mathrm{ML}}(N^\Omega, G) \neq R \right) = \Pr\left( \psi_{\mathrm{ML}}(N^\Omega, G) \neq R \mid R = R^{(\gamma)} \right).$$

The event "$\psi_{\mathrm{ML}}(N^\Omega, G) \neq R \mid R = R^{(\gamma)}$" happens only if there exists some rating matrix $X$ such that the likelihood of $X$ is greater than or equal to that of $R^{(\gamma)}$.

**Lemma 1.** *The negative log-likelihood of $X$ is equal to $\mathsf{L}(X) + c$ for $\mathsf{L}(X) := c_s e(A_X, B_X) - c_r \Pi(X)$ and some constant $c$ independent of the choice of $X$. Here, $\Pi(X)$ is the number of observed ratings which coincide with the corresponding ratings of $X$; $c_s := \log\left( \frac{(1-\beta)\alpha}{(1-\alpha)\beta} \right)$ and $c_r := \log\left( \frac{1-\theta}{\theta} \right)$ are positive constants. In particular, the likelihood of $X$ is greater than or equal to that of $R^{(\gamma)}$ if and only if $\mathsf{L}(X) \leq \mathsf{L}(R)$.*

*Proof:* See Appendix A-A. ∎

By Lemma 1 together with the union bound,

$$\Pr\left( \psi_{\mathrm{ML}}(N^\Omega, G) \neq R \mid R = R^{(\gamma)} \right) \leq \sum_{X \neq R^{(\gamma)}} \Pr\left( \mathsf{L}(X) \leq \mathsf{L}(R^{(\gamma)}) \right). \tag{2}$$

To enumerate all rating matrices different from $R^{(\gamma)}$, we define $\mathcal{X}(k, a_1, a_2, b_1, b_2)$ to be the class of rating matrices $X$'s such that (i) $|A_X \setminus A_{R^{(\gamma)}}| = |B_X \setminus B_{R^{(\gamma)}}| =: k$; (ii) $u_X$ differs from $u_{R^{(\gamma)}}$ at $a_1$ many coordinates among the first $(1-\gamma)m$ coordinates and at $a_2$ many coordinates among the next $\gamma m$ coordinates; and (iii) $v_X$ differs from $v_{R^{(\gamma)}}$ at $b_1$ many coordinates among the first $(1-\gamma)m$ coordinates and at $b_2$ many coordinates among the next $\gamma m$ coordinates. Note that if $X_1$ and $X_2$ belong to the same class, then $\Pr(\mathsf{L}(X_1) \leq \mathsf{L}(M)) = \Pr(\mathsf{L}(X_2) \leq \mathsf{L}(M))$ as the two events are statistically identical. Let $\mathcal{I}$ be the range of index, i.e., collection of tuples $(k, a_1, a_2, b_1, b_2) \neq (0,0,0,0,0)$ such that $0 \leq k \leq n/4$ and $0 \leq a_1, b_1 \leq (1-\gamma)m$ and $0 \leq a_2, b_2 \leq \gamma m$. Note that $k \leq n/4$ is sufficient as one can switch the role of $u_X$ and $v_X$.

With these notations, RHS of (2) is equal to

$$\sum_{z \in \mathcal{I}} \sum_{X \in \mathcal{X}(z)} \Pr\left( \mathsf{L}(X) \leq \mathsf{L}(R^{(\gamma)}) \right) = \sum_{z \in \mathcal{I}} |\mathcal{X}(z)| \Pr\left( \mathsf{L}(X_z) \leq \mathsf{L}(R^{(\gamma)}) \right), \tag{3}$$

where $X_z$ is an arbitrary rating matrix that belongs to $\mathcal{X}(z)$.

**Lemma 2.** *For $z = (k, a_1, a_2, b_1, b_2) \in \mathcal{I}$,*

$$\Pr\left( \mathsf{L}(X_z) \leq \mathsf{L}(R^{(\gamma)}) \right) = \Pr\left( c_s \sum_{i=1}^{2(n/2-k)k} (B_i - A_i) + c_r \sum_{i=1}^{\mathcal{D}_z} P_i(2\Theta_i - 1) \geq 0 \right),$$

where $\mathcal{D}_z := k \cdot \{a_1 + (\gamma m - a_2) + b_1 + (\gamma m - b_2)\} + \left(\frac{n}{2} - k\right) \cdot (a_1 + a_2 + b_1 + b_2)$, $A_i \overset{i.i.d.}{\sim} \mathrm{Bern}(\alpha)$, $B_i \overset{i.i.d.}{\sim} \mathrm{Bern}(\beta)$, $\{P_i\} \overset{i.i.d.}{\sim} \mathrm{Bern}(p)$, and $\{\Theta_i\} \overset{i.i.d.}{\sim} \mathrm{Bern}(\theta)$.

*Proof:* See Appendix A-B. ∎

Due to Lemma 2, $\Pr\left(\mathsf{L}(X_z) \leq \mathsf{L}(R^{(\gamma)})\right)$ is equal to a tail probability of a sum of random variables of form $c_s \sum_i (B_i - A_i) + c_r \sum_i P_i(2\Theta_i - 1)$. The following lemma provides a sharp bound on this tail probability.

**Lemma 3.** *For integers $K, L > 0$, let $\{A_i\}_{i=1}^K \overset{i.i.d.}{\sim} \mathrm{Bern}(\alpha)$, $\{B_i\}_{i=1}^K \overset{i.i.d.}{\sim} \mathrm{Bern}(\beta)$, $\{P_i\}_{i=1}^L \overset{i.i.d.}{\sim} \mathrm{Bern}(p)$ and $\{\Theta_i\}_{i=1}^L \overset{i.i.d.}{\sim} \mathrm{Bern}(\theta)$. Assume that $\alpha, \beta, p = o(1)$. Then, for any $\ell > 0$*

$$\Pr\left(\log\left(\frac{(1-\beta)\alpha}{(1-\alpha)\beta}\right)\sum_{i=1}^K (B_i - A_i) + \log\left(\frac{1-\theta}{\theta}\right)\sum_{i=1}^L P_i(2\Theta_i - 1) \geq -\ell\right) \leq e^{\frac{1}{2}\ell - (1+o(1))KI_s - (1+o(1))LI_r}. \quad (4)$$

*Proof.* See Appendix A-C. □

By Lemma 3[1] with $\ell = 0$, $K = 2\left(\frac{n}{2} - k\right)k$, and $L = \mathcal{D}_z$,[2]

$$\Pr(\mathsf{L}(X_z) \leq \mathsf{L}(R^{(\gamma)})) \leq e^{-2k(n/2-k)I_s - \mathcal{D}_z I_r}. \quad (5)$$

Hence, one can bound the RHS of (3) by

$$\sum_{z\in\mathcal{I}} |\mathcal{X}(z)| e^{-2\left(\frac{n}{2}-k\right)kI_s - \mathcal{D}_z I_r}. \quad (6)$$

For the rest of the proof, we will show that (6) converges to 0. Due to the condition $\frac{1}{2}nI_s + \gamma mI_r \geq (1+\epsilon)\log n$, either $mnI_r = \Omega(n\log n)$ or $n^2 I_s = \Omega(n\log n)$ holds. We only consider the former case and note that the other case can be proved similarly.

For a small constant $\delta \in (0, \min\{\gamma, 1-\gamma\})$, let

$$\mathcal{J} := \{(k, a_1, a_2, b_1, b_2) \in \mathcal{I} \ : \ a_1, a_2, b_1, b_2 < \delta m\} \text{ and } \mathcal{K} := \{(k, a_1, a_2, b_1, b_2) \in \mathcal{I} \ : \ k < \delta n\}$$

Then, we divide (6) into three partial sums according to subsets $\mathcal{I} \setminus \mathcal{J}$, $\mathcal{J} \setminus \mathcal{K}$, and $\mathcal{J} \cap \mathcal{K}$. Below, we show that each partial sum converges to zero.

1) $\mathcal{I} \setminus \mathcal{J}$: Among $a_1, a_2, b_1, b_2$, at least one is greater than equal to $\delta m$. Without loss of generality, assume $a_1 \geq \delta m$. From the definition of $\mathcal{D}_z$, one can see that $\mathcal{D}_z \geq \left(\frac{n}{2} - k\right)a_1 \geq \frac{\delta}{4}nm$ when $z \in \mathcal{I} \setminus \mathcal{J}$, which implies $\mathcal{D}_z = \Omega(nm)$. Hence, the summation for this case is upper bounded by

$$\sum_{z\in\mathcal{I}\setminus\mathcal{J}} |\mathcal{X}(z)| e^{-\mathcal{D}_z I_r} \leq e^{-\Omega(nm)I_r} \sum_{z\in\mathcal{I}\setminus\mathcal{J}} |\mathcal{X}(z)| \leq e^{-\Omega(nm)I_r} 2^n 2^{2m},$$

where the last inequality follows from the fact that the total number of rating matrices is bounded by $2^n 2^{2m}$. As $nmI_r = \Omega(n\log n + m\log m)$, the last term converges to zero.

2) $\mathcal{J} \setminus \mathcal{K}$: As $\mathcal{D}_z \geq k \cdot (\gamma m - a_2) \geq \delta n(\gamma - \delta)m = \Omega(nm)$ for $z \in \mathcal{J} \setminus \mathcal{K}$, a similar proof follows.

3) $\mathcal{J} \cap \mathcal{K}$: Due to the facts that $k\left(\frac{n}{2} - k\right) \geq k\left(\frac{1}{2} - \delta\right)n$ and $\mathcal{D}_z \geq k(\gamma m - a_2 + \gamma m - b_2) + \left(\frac{n}{2} - k\right)(a_1 + a_2 + b_1 + b_2) \geq 2k(\gamma - \delta)m + \left(\frac{1}{2} - \delta\right)n(a_1 + a_2 + b_1 + b_2)$,

$$e^{-2\left(\frac{n}{2}-k\right)kI_s - \mathcal{D}_z I_r} \leq \underbrace{e^{-2k\cdot\{(\frac{1}{2}-\delta)nI_s + (\gamma-\delta)mI_r\}}}_{(a)} \underbrace{e^{(\frac{1}{2}-\delta)n(a_1+a_2+b_1+b_2)I_r}}_{(b)}.$$

We estimate $(a)$ and $(b)$ separately. As for $(a)$, by taking $\delta$ sufficiently small (depending on $\epsilon$), the condition $\frac{1}{2}nI_s + \gamma mI_r \geq (1+\epsilon)\log n$ guarantees that $\left(\frac{1}{2} - \delta\right)nI_s + (\gamma - \delta)mI_r \geq \left(1 + \frac{\epsilon}{2}\right)\log n$. This implies $(a) \leq n^{-2(1+\frac{\epsilon}{2})k}$.

As for $(b)$, by taking $\delta$ sufficiently small (depending on $\epsilon$), the condition $\frac{1}{2}nI_r \geq (1+\epsilon)\log m$ ensures that $\left(\frac{1}{2} - \delta\right)nI_r \geq \left(1 + \frac{\epsilon}{2}\right)\log m$. Hence, we get $(b) \leq m^{-(1+\frac{\epsilon}{2})(a_1+a_2+b_1+b_2)}$.

Now, we turn to $|\mathcal{X}(z)|$. When $z = (k, a_1, a_2, b_1, b_2)$, one can easily check that $|\mathcal{X}(z)|$ is equal to $\binom{\frac{n}{2}}{k}^2 \binom{(1-\gamma)m}{a_1}\binom{(1-\gamma)m}{b_1}\binom{\gamma m}{a_2}\binom{\gamma m}{b_2}$, which can be easily upper bounded by $n^{2k} m^{a_1+a_2+b_1+b_2}$.

Combining above estimations, (6) is upper bounded by $\sum_{z\in\mathcal{I}} n^{-\epsilon k} m^{-\frac{\epsilon}{2}(a_1+a_2+b_1+b_2)}$, which converges to zero as $n \to \infty$ ($m \to \infty$ follows since $m = o(\log n)$).

## B. MLE Converse

Consider the (constrained) maximum likelihood estimator $(\psi_{\mathrm{ML}}|_{\mathcal{C}^{(\gamma)}})$ whose output is constrained in $\mathcal{C}^{(\gamma)}$. We will first prove that $\inf_{\psi} P_e^{(\gamma)}(\psi) \geq \Pr(\psi_{\mathrm{ML}}|_{\mathcal{C}^{(\gamma)}}(N^{\Omega}, G) \neq R \mid R = R^{(\gamma)})$. Suppose that $R'$ is a random matrix chosen uniformly random from $\mathcal{C}^{(\gamma)}$. Then, we have

$$
\begin{aligned}
\inf_{\psi} P_e^{(\gamma)}(\psi) &= \inf_{\psi} \max_{X \in \mathcal{C}^{(\gamma)}} \Pr(\psi(N^{\Omega}, G) \neq R \mid R = X) \\
&\overset{(a)}{\geq} \inf_{\psi} \Pr(\psi(N^{\Omega}, G) \neq R \mid R = R') \\
&\overset{(b)}{=} \inf_{\psi \,:\, \psi(N^{\Omega}, G) \in \mathcal{C}^{(\gamma)}} \Pr(\psi(N^{\Omega}, G) \neq R \mid R = R') \\
&\overset{(c)}{=} \Pr(\psi_{\mathrm{ML}}|_{\mathcal{C}^{(\gamma)}}(N^{\Omega}, G) \neq R \mid R = R') \\
&\overset{(d)}{=} \Pr(\psi_{\mathrm{ML}}|_{\mathcal{C}^{(\gamma)}}(N^{\Omega}, G) \neq R \mid R = R^{(\gamma)}),
\end{aligned}
$$

where $(a)$ holds since "max$\geq$mean"; $(b)$ follows since the $\psi(N^{\Omega}, G) \in \mathcal{C}^{(\gamma)}$ should be true for the optimal estimator; $(c)$ is due to the fact that ML is the optimal estimator under uniform prior; $(d)$ holds since $\Pr(\psi_{\mathrm{ML}}|_{\mathcal{C}^{(\gamma)}}(N^{\Omega}, G) \neq R \mid R = X)$ is the same quantity for any $X \in \mathcal{C}^{(\gamma)}$. Thus, we will show $\Pr(\psi_{\mathrm{ML}}|_{\mathcal{C}^{(\gamma)}}(N^{\Omega}, G) \neq R \mid R = R^{(\gamma)}) \nrightarrow 0$ if either $\frac{1}{2}nI_s + \gamma m I_r \leq (1 - \epsilon)\log n$ or $\frac{1}{2}nI_r \leq (1 - \epsilon)\log m$.

Let $S$ be the success event:

$$
S := \bigcap_{X \neq R^{(\gamma)}, \; X \in \mathcal{C}^{(\gamma)}} \left[ \mathsf{L}(X) > \mathsf{L}(R^{(\gamma)}) \right].
$$

Under the occurrence of $S^c$ (i.e., failure), there is $X \in \mathcal{C}^{(\gamma)}$ such that $X \neq R^{(\gamma)}$ and $\mathsf{L}(X) \leq \mathsf{L}(R^{(\gamma)})$, and hence $\Pr(\psi_{\mathrm{ML}}|_{\mathcal{C}^{(\gamma)}}(N^{\Omega}, G) \neq R \mid R = R^{(\gamma)}) \geq \frac{1}{2}$. This implies $\Pr(\psi(N^{\Omega}, G) \neq R \mid R = R^{(\gamma)}) \geq \frac{1}{2}\Pr(S^c)$. Hence, it is enough to prove $\Pr(S) \to 0$.

Owing to notation from the achievability proof, we have $\mathcal{X}(0, 0, 0, 1, 1) \subset \mathcal{C}^{(\gamma)}$ and $\mathcal{X}(1, 0, 0, 0, 0) \subset \mathcal{C}^{(\gamma)}$. Hence, the following two inequalities hold:

$$
\Pr(S) \leq \Pr\left( \bigcap_{X \in \mathcal{X}(0,0,0,1,1)} \left[ \mathsf{L}(X) > \mathsf{L}(R^{(\gamma)}) \right] \right) \tag{7}
$$

and

$$
\Pr(S) \leq \Pr\left( \bigcap_{X \in \mathcal{X}(1,0,0,0,0)} \left[ \mathsf{L}(X) > \mathsf{L}(R^{(\gamma)}) \right] \right). \tag{8}
$$

We finish the proof by showing (i) the RHS of (7) converges to 0 if $\frac{1}{2}nI_r \leq (1 - \epsilon)\log m$; and (ii) the RHS of (8) converges 0 if $\frac{1}{2}nI_s + \gamma m I_r \leq (1 - \epsilon)\log n$.

1) RHS(7) $\to 0$ if $\frac{1}{2}nI_r \leq (1 - \epsilon)\log m$: Suppose there exist $X_1 \in \mathcal{X}(0, 0, 0, 1, 0)$ and $X_2 \in \mathcal{X}(0, 0, 0, 0, 1)$ such that $\mathsf{L}(X_1) \leq \mathsf{L}(R^{(\gamma)})$ and $\mathsf{L}(X_2) \leq \mathsf{L}(R^{(\gamma)})$. Assume that $v_{X_1}$ differs from $v_{R^{(\gamma)}}$ at $i_1$-th coordinate and $v_{X_2}$ differs from $v_{R^{(\gamma)}}$ at $i_2$-th coordinate. Then, $\mathsf{L}(X_3) \leq \mathsf{L}(R^{(\gamma)})$ for $X_3 \in \mathcal{X}(0, 0, 0, 1, 1)$ whose $v_{X_3}$ differs from $v_{R^{(\gamma)}}$ at $i_1$ and $i_2$-th coordinates. Thus, if $\mathsf{L}(X) > \mathsf{L}(R^{(\gamma)})$ for any $X \in \mathcal{X}(0, 0, 0, 1, 1)$, then either one of the following holds: (i) $\mathsf{L}(X_1) > \mathsf{L}(R^{(\gamma)})$ for any $X_1 \in \mathcal{X}(0, 0, 0, 1, 0)$; or (ii) $\mathsf{L}(X_2) > \mathsf{L}(R^{(\gamma)})$ for any $X_2 \in \mathcal{X}(0, 0, 0, 0, 1)$.

Hence, the union bound yields that the right hand side of (7) is upper bounded by

$$
\Pr\left( \bigcap_{X_1 \in \mathcal{X}(0,0,0,1,0)} \left[ \mathsf{L}(X_1) > \mathsf{L}(R^{(\gamma)}) \right] \right) + \Pr\left( \bigcap_{X_2 \in \mathcal{X}(0,0,0,0,1)} \left[ \mathsf{L}(X_2) > \mathsf{L}(R^{(\gamma)}) \right] \right)
$$

Below, we only show that the first term of the RHS converges to zero; the proof for the second term is identical. By Lemma 2, when $X \in \mathcal{X}(0, 0, 0, 1, 0)$, $\Pr(\mathsf{L}(X) > \mathsf{L}(R)) = 1 - \Pr(\mathsf{L}(X) \leq \mathsf{L}(R))$ is equal to

$$
1 - \Pr\left( c_r \sum_{i=1}^{\frac{n}{2}} P_i(2\Theta_i - 1) \geq 0 \right).
$$

The following lemma, which shows the tightness of the bound presented in Lemma 3, provides the upper bound of the last term.

**Lemma 4.** *For integers $K, L > 0$, let $\{A_i\}_{i=1}^K \overset{i.i.d.}{\sim} \mathsf{Bern}(\alpha)$, $\{B_i\}_{i=1}^K \overset{i.i.d.}{\sim} \mathsf{Bern}(\beta)$, $\{P_i\}_{i=1}^L \overset{i.i.d.}{\sim} \mathsf{Bern}(p)$ and $\{\Theta_i\}_{i=1}^L \overset{i.i.d.}{\sim} \mathsf{Bern}(\theta)$. Assume that $\alpha, \beta, p = o(1)$ and $\max\left\{\sqrt{\alpha\beta}K, \ pL\right\} = \omega(1)$. Then, the following holds for sufficiently large $K$ if $\sqrt{\alpha\beta}K > pL$; sufficiently large $L$ otherwise:*

$$\Pr\left(\log\left(\frac{(1-\beta)\alpha}{(1-\alpha)\beta}\right)\sum_{i=1}^K (B_i - A_i) + \log\left(\frac{1-\theta}{\theta}\right)\sum_{i=1}^L P_i(2\Theta_i - 1) \geq 0\right) \geq \frac{1}{4}e^{-(1+o(1))KI_r - (1+o(1))LI_s}. \tag{9}$$

*Proof.* See Appendix A-D. □

By Lemma 4 with $K = 0$ and $L = \frac{n}{2}$, the last term is bounded by $1 - \frac{1}{4}e^{-\frac{n}{2}I_r} \leq e^{-\frac{1}{4}e^{-\frac{n}{2}I_r}}$. Note that the collection of events $\{[\mathsf{L}(X) > \mathsf{L}(R)]\}_{X \in \mathcal{X}(0,0,0,1,0)}$ is mutually independent as different events are tied to different column vectors of $N^\Omega$. Therefore, for a binary rating matrix matrix $X_0 \in \mathcal{X}(0,0,0,1,0)$, the RHS of (7) is equal to $\Pr\left(\mathsf{L}(X_0) > \mathsf{L}(R)\right)^{|\mathcal{X}(0,0,0,1,0)|}$, which is bounded by

$$\exp\left\{-\frac{1}{4}|\mathcal{X}(0,0,0,1,0)|e^{-(1+o(1))\frac{n}{2}I_r}\right\} = \exp\left\{-\frac{1}{4}\gamma m e^{-(1+o(1))\frac{n}{2}I_r}\right\}$$

$$= \exp\left\{-\frac{1}{4}\gamma e^{-(1+o(1))\frac{n}{2}I_r + \log m}\right\}. \tag{10}$$

The RHS of (10) goes to zero as $m \to \infty$ (which is true when $n \to \infty$ since $m = \omega(\log n)$) due to $\frac{1}{2}nI_r \leq (1-\epsilon)\log m$.

2) RHS(8) $\to 0$ if $\frac{1}{2}nI_s + \gamma m I_r \leq (1-\epsilon)\log n$: For this case, we make use of a combinatorial property of random graphs:

**Lemma 5.** *Suppose $\alpha = O\left(\frac{\log n}{n}\right)$, and consider the following procedure:*

1) *For $r = \frac{n}{\log^3 n}$, let $T := \{1, 2, \cdots, 2r\} \cup \left\{\frac{n}{2} + 1, \frac{n}{2} + 2, \cdots, \frac{n}{2} + 2r\right\}$.*
2) *Within $T$, we will delete every pair of two nodes which are adjacent.*
3) *Denote the remaining nodes by $U$.*

*With probability approaching $1$, the above procedure results in $|U| \geq 3\frac{n}{\log^3 n}$.*

*Proof.* See Appendix A-E. □

Let $\Delta$ be the event $\left[|U| \geq 3\frac{n}{\log^3 n}\right]$. We note that $\Delta$ is an event defined over edges within $T$. Conditioned on $\Delta$, one can find subsets $A_P \subset A_R$ and $A_Q \subset B_R$ such that (i) $|A_P| = |A_Q| = \frac{n}{\log^3 n}$ and (ii) there is no edge between nodes in $A_P \cup A_Q$. Without loss of generality, assume that $1 \in A_P$.

For a rating matrix $X$, Let $X^{(i)}$ be the rating matrix obtained from $X$ by replacing $i$th row with $v_X$ if $i \in A_X$; with $u_X$ otherwise. In other words, $A_{X^{(i)}} = A_X \triangle \{i\}$ and $B_{X^{(i)}} = B_X \triangle \{i\}$. As $\Pr(\Delta) = 1 - o(1)$, the RHS of (8) is upper bounded by

$$\Pr\left(\bigcap_{i \in A_P, \ j \in A_Q} \left[\mathsf{L}\left(\left(R^{(i)}\right)^{(j)}\right) > \mathsf{L}(R)\right] \ \middle|\ \Delta\right) \cdot (1 - o(1)). \tag{11}$$

**Lemma 6.** *Suppose that $\mathsf{L}(R^{(i)}) \leq \mathsf{L}(R)$ and $\mathsf{L}(R^{(j)}) \leq \mathsf{L}(R)$ hold for $i \in A_P$ and $j \in A_Q$. Then, conditioned on $\Delta$, $\mathsf{L}\left(\left(R^{(i)}\right)^{(j)}\right) \leq \mathsf{L}(R)$.*

*Proof.* See Appendix A-F. □

Due to Lemma 6, if $\mathsf{L}\left(\left(R^{(i)}\right)^{(j)}\right) > \mathsf{L}(R)$ for any $i \in A_P$ and $j \in A_Q$, then either one of the following holds: (i) $\mathsf{L}(R^{(i)}) > \mathsf{L}(R)$ for any $i \in A_P$; or (ii) $\mathsf{L}(R^{(j)}) > \mathsf{L}(R)$ for any $j \in A_Q$. Hence, it can be deduced from the union bound that the RHS of (11) is upper bounded by $\Pr\left(\bigcap_{i \in A_P}\left[\mathsf{L}\left(R^{(i)}\right) > \mathsf{L}(R)\right] \ \middle|\ \Delta\right) \cdot (1 - o(1)) + \Pr\left(\bigcap_{j \in A_Q}\left[\mathsf{L}\left(R^{(j)}\right) > \mathsf{L}(R)\right] \ \middle|\ \Delta\right) \cdot (1 - o(1))$. By symmetry, this upper bound is equal to $2\Pr\left(\bigcap_{i \in A_P}\left[\mathsf{L}\left(R^{(i)}\right) > \mathsf{L}(R)\right] \ \middle|\ \Delta\right) \cdot (1 - o(1))$. Due to the construction in Lemma 5, there are no edges between the nodes in $A_P$ conditioned on $\Delta$, implying that $\left\{\left[\mathsf{L}\left(R^{(i)}\right) > \mathsf{L}(R)\right]\right\}_{i \in A_P}$ is mutually independent. Thus, the last upper bound is equal to

$$2\Pr\left(\mathsf{L}\left(R^{(1)}\right) > \mathsf{L}(R) \ \middle|\ \Delta\right)^{|A_P|} \cdot (1 - o(1)). \tag{12}$$

This leads us to compute $\Pr\left(\mathsf{L}\left(R^{(1)}\right) > \mathsf{L}(R) \mid \Delta\right)$. As $\Pr(\Delta) \geq \frac{1}{2}$, $\Pr\left(\mathsf{L}\left(R^{(1)}\right) > \mathsf{L}(R) \mid \Delta\right) \leq 2\Pr\left(\mathsf{L}\left(R^{(1)}\right) > \mathsf{L}(R)\right)$. In light of Lemma 2, one obtains that $\Pr\left(\mathsf{L}\left(R^{(1)}\right) > \mathsf{L}(R)\right)$ is equal to

$$\Pr\left( c_s \sum_{j=1}^{n/2-1} (B_j - A_j) + c_s B_{n/2} + c_r \sum_{j=1}^{\gamma m} P_j \left(2\Theta_j - 1\right) < 0 \right). \tag{13}$$

Hence, by disregarding the term $c_s B_{n/2}$, this is bounded by $\Pr\left(c_s \sum_{j=1}^{n/2-1}(B_j - A_j) + c_r \sum_{j=1}^{\gamma m} P_j \left(2\Theta_j - 1\right) < 0\right)$. By Lemma 4 with $K = n/2 - 1$ and $L = \gamma m$, the last term is upper bounded by $2 - 2e^{-(1+o(1))(n/2-1)I_s - (1+o(1))\gamma m I_r}$, which in turn is bounded by

$$2\exp\left\{-e^{-(1+o(1))(n/2-1)I_s - (1+o(1))\gamma m I_r}\right\}.$$

Hence, (12) is bounded by

$$4\exp\left\{-\frac{n}{\log^3 n} e^{-(1+o(1))(n/2-1)I_s - (1+o(1))\gamma m I_r}\right\} \cdot (1 - o(1)), \tag{14}$$

which converges to zero as $n \to \infty$: due to the fact that $\frac{1}{2}nI_s + \gamma m I_r \leq (1 - \epsilon)\log n$, it follows that for sufficiently large $n$, $-(1+o(1))\left(n/2-1\right)I_s - (1+o(1))\gamma m I_r \geq -(1-\epsilon/2)\log n$, showing that (14) is bounded by $4\exp\left\{-\frac{n^{\epsilon/2}}{\log^3 n}\right\} \cdot (1 - o(1))$, which evidently converges to 0.

*C. Proof of Corollary 1*

We remark how the proof changes for Corollary 1. In the achievability proof, $\mathcal{I}$ becomes $\{(k, 0, 0, 0, 0) : k \leq n/4\}$; hence, the RHS of (6) goes to zero only with the condition $\frac{1}{2}nI_s + \gamma m I_r \geq (1+\epsilon)\log n$. The converse part is identical to the proof of RHS(8)$\to 0$.

## II. PROOF OF EFFICIENT ALGORITHM (THEOREM 2)

Since $I_s = \omega(\frac{1}{n})$, the spectral clustering is shown to output clusters with a vanishing error fraction [5, Theorem 6]. Hence, $A_1^{(0)}, A_2^{(0)}$—by switching them if necessary—coincides with $A_R, B_R$, except vanishing fractions. In the following subsections, we will show that (i) Stage 2 recovers $u_R$ and $v_R$ if $\frac{1}{2}nI_r \geq (1+\epsilon)\log m$ and (ii) Stage 3 recovers $A_R$ and $B_R$ if $\frac{1}{2}nI_s + \gamma m I_r \geq (1+\epsilon)\log n$.

*A. Stage 2 yields exact recovery of $u_R$ and $v_R$*

Under the success of Stage 1, we will show that $\Pr\left(u^{(1)} \neq u_R\right) = o(1)$; the proof for $u^{(2)}$ is similar. Let $R := A_1^{(0)} \setminus A_R$ and $\eta := |R|/n$. Due to Stage 1, $\lim_{n\to\infty} \eta = 0$ with high probability. For a fixed $1 \leq j \leq m$, we first estimate the probability $\Pr\left((u^{(1)})_j \neq (u_R)_j\right)$. Without loss of generality, assume $(u_R)_j = +1$. Since $(u^{(1)})_j \neq +1 \Leftrightarrow \sum_{i \in A_1^{(0)}} N_{ij}^{\Omega} \leq 0$, the probability is equal to $\Pr\left(\sum_{i \in A_1^{(0)}} N_{ij}^{\Omega} \leq 0\right)$, which is bounded by

$$\Pr\left( \sum_{i=1}^{(\frac{1}{2}-\eta)n} P_i \left(2\Theta_i - 1\right) \geq -\sum_{i=1}^{\eta n} P_i' \right), \tag{15}$$

where $\{P_i\}, \{P_i'\} \overset{\text{i.i.d.}}{\sim} \text{Bern}(p)$ and $\{\Theta_i\} \overset{\text{i.i.d.}}{\sim} \text{Bern}(\theta)$. Indeed, this bound follows since $-\sum_{i \in A_1^{(0)} \setminus R} N_{ij}^{\Omega} \sim \sum_{i=1}^{(\frac{1}{2}-\eta)n} P_i \left(2\Theta_i - 1\right)$ and $\sum_{i \in R} N_{ij}^{\Omega} \geq -\sum_{i \in R} \left|N_{ij}^{\Omega}\right| \sim -\sum_{i=1}^{\eta n} P_i'$.

We now estimate (15). We first show that $-\sum_{i=1}^{\eta n} P_i'$ in (15) using the following large deviation result.

**Lemma 7.** *Suppose that $X \sim \text{Binom}(\epsilon n, p)$ for some $0 < \epsilon < 1$ and $0 < p < 1/2$. Then for any $k \geq 2e$, one has $\Pr\left(X \geq \frac{knp}{\log \frac{1}{\epsilon}}\right) \leq 2\exp\left(-\frac{knp}{2}\right)$.*

*Proof.* See Appendix A-G. □

By Lemma 7, for $c > 4e$, $\Pr\left(\sum_{i=1}^{\eta n} P_i' \geq \frac{cnp}{\log \frac{1}{\eta}}\right) \leq 2\exp\left(-\frac{cnp}{2}\right)$. As $np = \Omega(\log m)$, by taking $c$ sufficiently large, we obtain $\Pr\left(\sum_{i=1}^{\eta n} P_i' \geq \frac{cnp}{\log \frac{1}{\eta}}\right) = o(m^{-1})$. Thus, (15) is bounded by $\Pr\left(\sum_{i=0}^{(\frac{1}{2}-\eta)n} P_i \left(2\Theta_i - 1\right) \geq -\frac{cnp}{\log \frac{1}{\eta}}\right) + o(m^{-1})$. By Lemma 3 with $K = 0$, $L = \left(\frac{1}{2} - \eta\right)n$, and $\ell = \frac{cnp}{\log \frac{1}{\eta}}$, this is then upper bounded by

$$\exp\left( \frac{1}{2}\log\left(\frac{1-\theta}{\theta}\right) \frac{c}{\log \frac{1}{\eta}} np - (1+o(1))\left(\frac{1}{2}-\eta\right)nI_r \right) + o(m^{-1}). \tag{16}$$

Since $np = \Theta(nI_r)$ and $\lim_{\eta \to 0+} \frac{1}{\log \frac{1}{\eta}} = 0$, the first term in the exponent is negligible compared to the second term in the exponent as $\eta$ tends to zero. Moreover, due to $\frac{1}{2}nI_r > (1+\epsilon)\log m$, for sufficiently small $\eta > 0$, the second term in the exponent is lower bounded by $(1+\epsilon/2)\log m$. Recalling the fact $\lim_{n \to \infty} \eta = 0$ with high probability, we can conclude that (16) is upper bounded by $\exp\left(-(1+\epsilon/2)\log m\right) + o(m^{-1})$, which is $o(m^{-1})$. Now, the proof is completed after taking union bound over all $j$'s.

### B. Stage 3 achieves exact recovery of $A_R$ and $B_R$

We assume that $u_R$ and $v_R$ are exactly recovered due to the analysis of Stage 2. For simplicity, we will focus on the case where $\alpha = O\left(\frac{\log n}{n}\right)$ and $p = O\left(\frac{\log n}{m} + \frac{\log m}{n}\right)$. The remaining cases can be dealt with similarly.

Note that when we choose $c_1' = \log\left(\frac{\alpha(1-\beta)}{\beta(1-\alpha)}\right)/\log\left(\frac{1-\theta}{\theta}\right)$ instead of $\log\left(\frac{\hat\alpha(1-\hat\beta)}{\hat\beta(1-\hat\alpha)}\right)/\log\left(\frac{1-\hat\theta}{\hat\theta}\right)$, the non-agnostic refinement rule can be written as follows: put user $i$ to $A_1^{(t)}$ if $\mathsf{L}(i; A_1^{(t-1)}, A_2^{(t-1)}) < 0$; $A_2^{(t)}$ otherwise, where $\mathsf{L}(i; A, B)$ is defined as $\mathsf{L}(i; A, B) := c_s\{e(\{i\}, A) - e(\{i\}, B)\} - c_r\{\Pi_i(u_R) - \Pi_i(v_R)\}$ for clusters $A$, $B$. Due to Lemma 1, this is indeed the local likelihood comparison, which updates each user's affiliation by the one giving better likelihood.

Next, consider the refinement rule of interest, defined by $c_1' = \log\left(\frac{\hat\alpha(1-\hat\beta)}{\hat\beta(1-\hat\alpha)}\right)/\log\left(\frac{1-\hat\theta}{\hat\theta}\right)$. This refinement rule can be written with $\widehat{\mathsf{L}}(i; A, B) := \hat c_s\{e(\{i\}, A) - e(\{i\}, B)\} - \hat c_r\{\Pi_i(u_R) - \Pi_i(v_R)\}$ instead of $\mathsf{L}(i; A, B)$, where $\hat c_s := \log\left(\frac{\hat\alpha(1-\hat\beta)}{\hat\beta(1-\hat\alpha)}\right)$ and $\hat c_r := \log\left(\frac{1-\hat\theta}{\hat\theta}\right)$. It follows from the analyses of the previous stages that the rating matrix $R^{(0)}$ constructed based on the outputs of Stage 1 and 2 almost coincides with the ground truth, and hence, $\hat c_s$ and $\hat c_r$ will provide accurate estimations of $c_s$ and $c_r$, respectively. This implies that $\widehat{\mathsf{L}}(i; A, B)$ is close to $\mathsf{L}(i; A, B)$, meaning the refinement rule of interest is also roughly equal to the local likelihood comparison.

For $\delta \in [n^{-1}, 1/2)$, let $\mathcal{Z}_\delta := \{(A, B) : A \cup B = [n], A \cap B = \emptyset, |A \triangle A_R| = |B \triangle B_R| < \delta n\}$. We will show that there exists a constant $\delta_0 > 0$ such that if $\delta < \delta_0$, the following statement holds with probability $1 - O(n^{-\epsilon/2})$: whenever $(A, B)$ belongs to $\mathcal{Z}_\delta$, the result of single step of iteration belongs to $\mathcal{Z}_{\delta/2}$. Then, $T = \frac{\log(\delta_0 n)}{\log 2}$ many iterations will suffice to guarantee exact recovery.

*Remark* 1. One might wonder why we prove that the refinement corrects every almost-correct rating in $\mathcal{Z}_\delta$ rather than just $R^{(0)}$, the output of Stage 1 and 2. This is because of subtle dependencies, which precludes the error analysis developed in Sec. I-A. More specifically, the entries of $R^{(0)}$ are no longer independent of each other, and hence, the error event cannot be expressed as in Lemma 2. This technique, called uniform analysis, is inspired by [3].

We first show that $(A_R, B_R)$ is a strict local optimum of the likelihood function under the condition $\frac{1}{2}nI_s + \gamma mI_r \geq (1+\epsilon)\log n$, i.e., the ground truth clusters are unaltered by an iteration of local refinement based on $\mathsf{L}$.

**Lemma 8.** *Suppose $\frac{1}{2}nI_s + \gamma mI_r \geq (1+\epsilon)\log n$. Then, there exists a small constant $\tau > 0$ such that the following holds with probability $1 - O(n^{-\epsilon/2})$: $\mathsf{L}(i; A_R, B_R) < -\tau \log n$ if $i \in A_R$; $\mathsf{L}(i; A_R, B_R) > \tau \log n$ otherwise.*

*Proof.* See Appendix A-H. □

Let $(A, B) \in \mathcal{Z}_\delta$. Due to Lemma 8, to show that the affiliation of node $i$ is the same as that of the ground truth after an iteration of refinement, it suffices to show:

$$\left|\widehat{\mathsf{L}}(i; A, B) - \mathsf{L}(i; A_R, B_R)\right| \leq \tau \log n. \tag{17}$$

Indeed, if one can show (17), it follows that

$$\begin{cases} \widehat{\mathsf{L}}(i; A, B) \leq \mathsf{L}(i; A_R, B_R) + \left|\widehat{\mathsf{L}}(i; A, B) - \mathsf{L}(i; A_R, B_R)\right| < 0 & \text{for } i \in A_R; \\ \widehat{\mathsf{L}}(i; A, B) \geq \mathsf{L}(i; A_R, B_R) - \left|\widehat{\mathsf{L}}(i; A, B) - \mathsf{L}(i; A_R, B_R)\right| > 0 & \text{otherwise.} \end{cases} \tag{18}$$

**Lemma 9.** *Suppose $n\alpha = \Theta(\log n)$. Then, for any constant $\tau > 0$, there exists $\delta_0 < 1/2$ such that if $\delta < \delta_0$, the following holds with probability $1 - O(n^{-1})$: for any $(A, B) \in \mathcal{Z}_\delta$, $|\mathsf{L}(i; A, B) - \mathsf{L}(i; A_R, B_R)| \leq \frac{\tau}{2}\log n$, for all except $\frac{\delta}{2}$ many nodes $i$'s.*

*Proof.* See Appendix A-I. □

**Lemma 10.** *Suppose $n\alpha = \Theta(\log n)$, $p = \Theta\left(\frac{\log n}{m} + \frac{\log m}{n}\right)$ and $m = O(n)$. Then, for any constant $\tau > 0$, the following holds with probability approaching 1: for any $A, B \subset [n]$, and $i \in [n]$, $\left|\widehat{\mathsf{L}}(i; A, B) - \mathsf{L}(i; A, B)\right| \leq \frac{\tau}{2}\log n$.*

*Proof.* See Appendix A-J. □

The above two lemmas together with the triangle inequality conclude that the output of refinement belongs to $\mathcal{Z}_{\delta/2}$.

## Footnotes

[1]Note that $\alpha, \beta, p = o(1)$ due to the following reasons: (i) $\alpha, \beta = o(1)$ as we focus on the case $\alpha, \beta = O\left(\frac{\log n}{n}\right)$; (ii) since we focus on the case $p = O\left(\frac{\log n}{m} + \frac{\log m}{n}\right)$, the conditions $m = \omega(\log n)$ and $\log m = o(n)$ ensure $p = o(1)$.

[2]Indeed, the upper bound due to Lemma 3 should read $e^{-(1+o(1))2k(n/2-k)I_s - (1+o(1))\mathcal{D}_z I_r}$ instead of (5). On the other hand, we will drop the $(1 + o(1))$'s for simplicity.

[3] Indeed, one can check that $\frac{1}{2} = \arg\min_{t>0}\mathbb{M}_{\log\left(\frac{(1-\beta)\alpha}{(1-\alpha)\beta}\right)(B-A)}(t) = \arg\min_{t>0}\mathbb{M}_{\log\left(\frac{1-\theta}{\theta}\right)P(2\Theta-1)}(t)$, meaning $(t = 1/2)$ is the optimal choice.

[4]Note that $\frac{2\log n}{n\alpha}$ is a constant as we assumed $\alpha = \Theta(\frac{\log n}{n})$ at the beginning of Sec. II-B.

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

## APPENDIX A
## PROOF OF LEMMAS

### A. Proof of Lemma 1

In this section, we compute the likelihood of a binary rating matrix matrix $X$ given $N^\Omega$ and $G$. Direct calculations yield

$$\Pr\left(N^\Omega \mid R = X\right) = (1-p)^{|\Omega|} p^{nm-|\Omega|} \theta^{|\Omega|-\Pi(X)}(1-\theta)^{\Pi(X)}$$

and

$$\Pr\left(G \mid R = X\right) = \alpha^{|E|-e(A_X, B_X)}(1-\alpha)^{2\binom{n/2}{2}-\{|E|-e(A_X,B_X)\}} \beta^{e(A_X,B_X)}(1-\beta)^{\left(\frac{n}{2}\right)^2-e(A_X,B_X)}.$$

By taking log on the both sides, we obtain $\log \Pr\left(N^\Omega \mid R = X\right) = \Pi(X)\log\left(\frac{1-\theta}{\theta}\right) + c$ and $\log \Pr\left(G \mid R = X\right) = e(A_X, B_X)\log\left(\frac{(1-\alpha)\beta}{(1-\beta)\alpha}\right) + c'$, where $c := \log\left\{(1-p)^{|\Omega|}p^{nm-|\Omega|}\theta^{|\Omega|}\right\}$ and $c' := \log\left\{\alpha^{|E|}(1-\alpha)^{2\binom{n/2}{2}-|E|}(1-\beta)^{\left(\frac{n}{2}\right)^2}\right\}$ are constants independent of the choice of $X$. Hence, the negative log-likelihood of $X$ is equal to $-\log\left(\frac{1-\theta}{\theta}\right)\Pi(X) + \log\left(\frac{(1-\beta)\alpha}{(1-\alpha)\beta}\right)e(A_X, B_X) + c + c'$.

### B. Proof of Lemma 2

Let $z = (k, a_1, a_2, b_1, b_2) \in \mathcal{I}$. Due to the definition of L,

$$\Pr\left(\mathsf{L}(X_z) \leq \mathsf{L}(R^{(\gamma)})\right) = \Pr\left(c_s\left\{e(A_{R^{(\gamma)}}, B_{R^{(\gamma)}}) - e(A_{X_z}, B_{X_z})\right\} - c_r\left\{\Pi(R^{(\gamma)}) - \Pi(X_z)\right\} \geq 0\right).$$

First, it is straightforward from the facts $|A_{X_z} \setminus A_{R^{(\gamma)}}| = |B_{X_z} \setminus B_{R^{(\gamma)}}| = k$ and $|A_{X_z} \cap A_{R^{(\gamma)}}| = |B_{X_z} \cap B_{R^{(\gamma)}}| = \frac{n}{2} - k$ that $e(A_{R^{(\gamma)}}, B_{R^{(\gamma)}}) - e(A_{X_z}, B_{X_z}) = \sum_{i=1}^{2(n/2-k)k}(B_i - A_i)$ (e.g. see Sec. 6 in [1]). Next, we compute $\Pi(R^{(\gamma)}) - \Pi(X_z)$. Note that $\Pi(R^{(\gamma)}) - \Pi(X_z) = \sum_{(i,j)\in\Omega}\left[\mathbb{I}\left\{N_{ij}^\Omega = (R^{(\gamma)})_{ij}\right\} - \mathbb{I}\left\{N_{ij}^\Omega = (X_z)_{ij}\right\}\right]$. Since $\mathbb{I}\left\{N_{ij}^\Omega = (R^{(\gamma)})_{ij}\right\} = \mathbb{I}\left\{N_{ij}^\Omega = (X_z)_{ij}\right\}$ whenever $(R^{(\gamma)})_{ij} = (X_z)_{ij}$, the last term is equal to $\sum_{\substack{(i,j)\in\Omega \,:\\(R^{(\gamma)})_{ij}\neq(X_z)_{ij}}}\left[\mathbb{I}\left\{N_{ij}^\Omega = (R^{(\gamma)})_{ij}\right\} - \mathbb{I}\left\{N_{ij}^\Omega = (X_z)_{ij}\right\}\right]$, which can be written as

$$\sum_{\ell=1}^{\left|\{(i,j)\,:\,(R^{(\gamma)})_{ij}\neq(X_z)_{ij}\}\right|}[P_\ell(1-\Theta_\ell) - P_\ell\Theta_\ell]$$

$$= \sum_{\ell=1}^{\left|\{(i,j)\,:\,(R^{(\gamma)})_{ij}\neq(X_z)_{ij}\}\right|}[P_\ell(1-2\Theta_\ell)].$$

A simple calculation yields $\left|\{(i,j)\,:\,(R^{(\gamma)})_{ij}\neq(X_z)_{ij}\}\right| = \mathcal{D}_z$. Hence, $\Pi(R^{(\gamma)}) - \Pi(X_z) = \sum_{i=1}^{\mathcal{D}_z}[P_i(1-2\Theta_i)]$.

### C. Proof of Lemma 3

Let $Z := \log\left(\frac{(1-\beta)\alpha}{(1-\alpha)\beta}\right)\sum_{i=1}^{K}(B_i-A_i) + \log\left(\frac{1-\theta}{\theta}\right)\sum_{i=1}^{L}P_i(2\Theta_i-1) + \ell$, $\mathbb{M}_1(t) := \mathbb{M}_{\log\left(\frac{(1-\beta)\alpha}{(1-\alpha)\beta}\right)(B_1-A_1)}(t)$, and $\mathbb{M}_2(t) := \mathbb{M}_{\log\left(\frac{1-\theta}{\theta}\right)P_1(2\Theta_1-1)}(t)$.

By Chernoff bound [4],

$$\Pr\left(Z > 0\right) \leq \inf_{t>0}\mathbb{M}_Z(t) \leq e^{\frac{1}{2}\ell}\mathbb{M}_1^K\left(\frac{1}{2}\right)\mathbb{M}_2^L\left(\frac{1}{2}\right) = e^{\frac{1}{2}\ell - KJ_1 - LJ_2},$$

where $J_1 := -\log\mathbb{M}_1(1/2)$ and $J_2 := -\log\mathbb{M}_2(1/2)$.[3]

We show that $J_1 = (1 + o(1))I_s$ and $J_2 = (1 + o(1))I_r$. A simple calculations yield

$$\mathbb{M}_1\left(\frac{1}{2}\right) = \left(\sqrt{\alpha\beta} + \sqrt{(1-\alpha)(1-\beta)}\right)^2; \tag{19}$$

$$\mathbb{M}_2\left(\frac{1}{2}\right) = 2p\sqrt{(1-\theta)\theta} + 1 - p. \tag{20}$$

Thus,

$$J_1 = -2\log\left(\sqrt{\alpha\beta} + \sqrt{(1-\alpha)(1-\beta)}\right) \overset{(a)}{=} -2\log\left\{\sqrt{\alpha\beta} + \left(1 - \frac{1}{2}\alpha + O(\alpha^2)\right)\left(1 - \frac{1}{2}\beta + O(\beta^2)\right)\right\}$$

$$= -2\log\left(1 + \sqrt{\alpha\beta} + -\frac{1}{2}\alpha - \frac{1}{2}\beta + O(\alpha^2 + \beta^2)\right) \overset{(b)}{=} -2\left(\sqrt{\alpha\beta} - \frac{1}{2}\alpha - \frac{1}{2}\beta + O(\alpha^2 + \beta^2)\right)$$

$$= (\sqrt{\alpha} - \sqrt{\beta})^2 + O(\alpha^2 + \beta^2) \overset{(c)}{=} (1 + o(1))I_s;$$

$$J_2 = -\log\left(2p\sqrt{(1-\theta)\theta} + 1 - p\right) \overset{(d)}{=} p(1 - \sqrt{(1-\theta)\theta}) + O(p^2) = p(\sqrt{1-\theta} - \sqrt{\theta})^2 + O(p^2) \overset{(e)}{=} (1 + o(1))I_r,$$

where $(a)$ follows from the fact that $\sqrt{1-x} = 1 - \frac{1}{2}x + O(x^2)$ as $x \to 0$; $(b)$ and $(d)$ follow from the fact that $\log(1+x) = x + O(x^2)$ as $x \to 0$; $(c)$ follows since $\alpha, \beta = o(1)$; $(e)$ follows as $p = o(1)$.

### D. Proof of Lemma 4

We adopt the techniques from [7] and [6]. Let $Z := \log\left(\frac{(1-\beta)\alpha}{(1-\alpha)\beta}\right)\sum_{i=1}^{K}(B_i - A_i) + \log\left(\frac{1-\theta}{\theta}\right)\sum_{i=1}^{L}P_i(2\Theta_i - 1)$, $X_i := \log\left(\frac{(1-\beta)\alpha}{(1-\alpha)\beta}\right)(B_i - A_i)$, for $i = 1, \ldots, K$, and $Y_j := \log\left(\frac{1-\theta}{\theta}\right)P_j(2\Theta_j - 1)$, for $i = 1, \ldots, L$.

Then, for a positive quantity $\xi$ (to be determined later),

$$\Pr(Z > 0) = \sum_{\substack{(x_1,\ldots,x_K,y_1,\ldots,y_L):\\ \sum_i x_i + \sum_j y_j > 0}} \prod_{i=1}^{K} p_{X_1}(x_i) \prod_{j=1}^{L} p_{Y_1}(y_j) \geq \sum_{\substack{(x_1,\ldots,x_K,y_1,\ldots,y_L):\\ \sum_i x_i + \sum_j y_j \in (0,\xi)}} \prod_{i=1}^{K} p_{X_1}(x_i) \prod_{j=1}^{L} p_{Y_1}(y_j)$$

$$\overset{(a)}{\geq} \frac{(\mathbb{M}_{X_1}(1/2))^K (\mathbb{M}_{Y_1}(1/2))^L}{e^{\frac{1}{2}\xi}} \sum_{\substack{(x_1,\ldots,x_K,y_1,\ldots,y_L):\\ \sum_i x_i + \sum_j y_j \in (0,\xi)}} \prod_{i=1}^{K} \frac{e^{\frac{1}{2}x_i} p_{X_1}(x_i)}{\mathbb{M}_{X_1}(1/2)} \prod_{j=1}^{L} \frac{e^{\frac{1}{2}y_i} p_{Y_1}(y_i)}{\mathbb{M}_{Y_1}(1/2)}$$

$$\overset{(b)}{=} e^{-KJ_1 - LJ_2 - \frac{1}{2}\xi} \sum_{\substack{(x_1,\ldots,x_K,y_1,\ldots,y_L):\\ \sum_i x_i + \sum_j y_j \in (0,\xi)}} \prod_{i=1}^{K} \frac{e^{\frac{1}{2}x_i} p_{X_1}(x_i)}{\mathbb{M}_{X_1}(1/2)} \prod_{j=1}^{L} \frac{e^{\frac{1}{2}y_i} p_{Y_1}(y_i)}{\mathbb{M}_{Y_1}(1/2)}$$

$$\overset{(c)}{=} e^{-KJ_1 - LJ_2 - \frac{1}{2}\xi} \underbrace{\Pr\left(0 < \sum_{i=1}^{K} V_i + \sum_{j=1}^{L} W_j < \xi\right)}_{(R1)},$$

where $(a)$ follows from the fact that $e^{\frac{1}{2}\xi} \geq e^{\frac{1}{2}(\sum_i x_i + \sum_j y_j)}$ when $\sum_i x_i + \sum_j y_j < \xi$; we define $J_1 := -\log\mathbb{M}_{X_1}(1/2)$ and $J_2 = -\log\mathbb{M}_{Y_1}(1/2)$ at $(b)$; at $(c)$, we define $V_1, \ldots, V_K$ to be i.i.d. random variables with distribution function $p_{V_1}(w) = \frac{e^{\frac{1}{2}w} p_{X_1}(w)}{\mathbb{M}_{X_1}(1/2)}$, and $W_1, \ldots, W_L$ to be i.i.d. random variables with distribution $p_{W_1}(w) = \frac{e^{\frac{1}{2}w} p_{Y_1}(w)}{\mathbb{M}_{Y_1}(1/2)}$. Here, the proof of Lemma 3 (see Appendix A-C) guarantees that $J_1 = (1 + o(1))I_s$ and $J_2 = (1 + o(1))I_r$.

Thus, it is enough to choose $\xi$ so that the followings holds for sufficiently large $K$ if $\sqrt{\alpha\beta}K > pL$; sufficiently large $L$ otherwise: (i) $e^{-KJ_1 - LJ_2 - \frac{1}{2}\xi} = (1 + o(1))e^{-KJ_1 - LJ_2}$; (ii) $(R1) > \frac{1}{4}$.

We will focus on the case of $pL \geq \sqrt{\alpha\beta}K$; the other case follows similarly. Take $\xi = (pL)^{3/4}$. Then, (i) is true due to the fact that $pL = \omega(1)$. The following lemma verifies (ii):

**Lemma 11.** As $L \to \infty$, we have $\Pr\left(\sum_{i=1}^{K} V_i + \sum_{j=1}^{L} W_j \geq (pL)^{3/4}\right) \to 0$.

*Proof.* See Appendix A-K. $\square$

Using the fact that $V_i$'s and $W_j$'s are mean zero random variables together with Lemma 11, we obtain

$$(R1) = \frac{1}{2} - \Pr\left(\sum_{i=1}^{K} V_i + \sum_{j=1}^{L} W_j \geq (pL)^{3/4}\right) \to \frac{1}{2}.$$

Hence, for sufficiently large $L$, we obtain $(R1) \geq \frac{1}{4}$.

## E. Proof of Lemma 5

The proof is based on the deletion technique (alteration technique) [2]. Let $\mathcal{F}$ be the number of nodes that are deleted in the step 2 of the procedure. Note that $\mathcal{F}$ is statistically dominated by $D := 2\sum_{i=1}^{\binom{4r}{2}} A_i$, where $A_i \overset{\text{i.i.d.}}{\sim} \text{Bern}(\alpha)$.

By definition, $\mathbb{E}[D] = \binom{4r}{2} 2\alpha \le (4r)^2 \alpha$. Thus, by Markov's inequality,

$$\Pr\left(D \ge \frac{r}{\log n}\right) \le \frac{\mathbb{E}[D]}{\frac{r}{\log n}} \le 16r\alpha \log n = O\left(\frac{r\log^2 n}{n}\right) = O\left(\frac{1}{\log n}\right).$$

This concludes that $D$ is almost surely of size $o(1) \cdot r$, which implies $|U|$ is larger than $3r$.

## F. Proof of Lemma 6

For fixed $i \in A_P$ and $j \in A_Q$, it suffices to show that conditioned on $\Delta$, $\mathsf{L}\left((R^{(i)})^{(j)}\right) - \mathsf{L}(R) = \left(\mathsf{L}(R^{(i)}) - \mathsf{L}(R)\right) + \left(\mathsf{L}(R^{(j)}) - \mathsf{L}(R)\right)$. By definition, showing the above is equivalent to checking the following two equalities:

$$\Pi((R^{(i)})^{(j)}) - \Pi(R) = \left(\Pi(R^{(i)}) - \Pi(R)\right) + \left(\Pi(R^{(j)}) - \Pi(R)\right) \quad \text{and} \tag{21}$$

$$e(A_{(R^{(i)})^{(j)}}, B_{(R^{(i)})^{(j)}}) - e(A_R, B_R) = (e(A_{R^{(i)}}, B_{R^{(i)}}) - e(A_R, B_R)) + (e(A_{R^{(j)}}, B_{R^{(j)}}) - e(A_R, B_R)). \tag{22}$$

First, (21) is straightforward from the definition of $\Pi(\cdot)$. Next, observe that

$$e(A_{(R^{(i)})^{(j)}}), B_{(R^{(i)})^{(j)}}) - e(A_R, B_R) = e(i, A_R) + e(j, B_R) - e(i, B_R \triangle \{j\}) - e(j, A_R \triangle \{i\}),$$
$$e(A_{R^{(i)}}, B_{R^{(i)}}) - e(A_R, B_R) = e(i, A_R) - e(j, B_R \triangle \{j\}) - \mathbb{I}\{\{ij\} \text{ is an edge}\}, \quad \text{and}$$
$$e(A_{R^{(j)}}, B_{R^{(j)}}) - e(A_R, B_R) = e(j, B_R) - e(j, A_R \triangle \{i\}) - \mathbb{I}\{\{ij\} \text{ is an edge}\}.$$

Since $i$ and $j$ are not adjacent conditioned on $\Delta$, (22) follows.

## G. Proof of Lemma 7

By definition, we have

$$\Pr\left(X \ge \frac{knp}{\log \frac{1}{\epsilon}}\right) = \sum_{i \ge \frac{np}{\log \frac{1}{\epsilon}}} \Pr(X = i) = \sum_{i \ge \frac{knp}{\log \frac{1}{\epsilon}}} \binom{\epsilon n}{i} p^i (1-p)^{\epsilon n - i}.$$

Due to the estimate $\binom{a}{b} \le \left(\frac{ea}{b}\right)^b$, the last term can be bounded as

$$e^{-\epsilon n p} \sum_{i \ge \frac{knp}{\log \frac{1}{\epsilon}}} \left(\frac{e\epsilon n}{i}\right)^i p^i (1-p)^{-i} \le \sum_{i \ge \frac{knp}{\log \frac{1}{\epsilon}}} \left(\frac{2e\epsilon np}{i}\right)^i \le \sum_{i \ge \frac{knp}{\log \frac{1}{\epsilon}}} \left(\frac{2e\epsilon np}{\frac{knp}{\log \frac{1}{\epsilon}}}\right)^i = \sum_{i \ge \frac{knp}{\log \frac{1}{\epsilon}}} \left(\frac{2e\epsilon \log \frac{1}{\epsilon}}{k}\right)^i$$

$$\le \sum_{i \ge \frac{knp}{\log \frac{1}{\epsilon}}} \left(\frac{2e\sqrt{\epsilon}}{k}\right)^i \le 2\left(\frac{2e\sqrt{\epsilon}}{k}\right)^{\frac{knp}{\log \frac{1}{\epsilon}}} = 2\exp\left(-\log\left(\frac{k}{2e\sqrt{\epsilon}}\right)\frac{knp}{\log \frac{1}{\epsilon}}\right) \le 2\exp\left(-\frac{knp}{2}\right),$$

where the first inequality follows since $1 - p \ge \frac{1}{2}$; the second inequality is due to $i \ge \frac{knp}{\log \frac{1}{\epsilon}}$; the third inequality holds since $\epsilon \log \frac{1}{\epsilon} \le \sqrt{\epsilon}$ is true for any $0 < \epsilon \le 1$; the fourth inequality is true due to the inequality $\sum_{i \ge b} a^i \le \frac{a^b}{1-a} \le 2a^b$ when $a \in (0, 1/2)$; the last inequality holds since $k \ge 2e$.

## H. Proof of Lemma 8

Let $\tau > 0$ be a constant to be chosen later. Without loss of generality, we assume $i \in A_R$. Note that it is enough to show

$$\Pr\left(\mathsf{L}(i; A_R, B_R) \ge -\tau \log n\right) = O(n^{-1-\epsilon/2})$$

since the lemma will follow after taking the union bound.

In light of Lemma 2, $\mathsf{L}(1; A_R, B_R) = \log\left(\frac{(1-\beta)\alpha}{(1-\alpha)\beta}\right) \sum_{j=1}^{n/2-1}(B_j - A_j) + B_{n/2} + \log\left(\frac{1-\theta}{\theta}\right) \sum_{j=1}^{\gamma m} P_j(2\Theta_j - 1)$.

Thus, $\Pr\left(\mathsf{L}(1; A_R, B_R) \ge \tau \log n\right) = \Pr\left(\log\left(\frac{(1-\beta)\alpha}{(1-\alpha)\beta}\right) \sum_{j=1}^{n/2-1}(B_j - A_j) + B_{n/2} + \log\left(\frac{1-\theta}{\theta}\right) \sum_{j=1}^{\gamma m} P_j(2\Theta_j - 1) \ge -\tau \log n\right)$

$\le \Pr\left(\log\left(\frac{(1-\beta)\alpha}{(1-\alpha)\beta}\right) \sum_{j=1}^{n/2-1}(B_j - A_j) + \log\left(\frac{1-\theta}{\theta}\right) \sum_{j=1}^{\gamma m} P_j(2\Theta_j - 1) \ge -\tau \log n - 1\right)$

$\overset{(a)}{\le} e^{\frac{1}{2}\tau \log n + \frac{1}{2} - \left(\frac{n}{2}-1\right)J_1 - \gamma m J_2} \overset{(b)}{\le} e^{\frac{1}{2}\tau \log n - (1+\epsilon)(1+o(1))\log n}$, where $(a)$ is due to Lemma 3; $(b)$ follows from the fact that $nI_s + 2\gamma m I_r > (1+\epsilon)\log n$. Hence, by taking $\tau$ sufficiently small, we complete the proof.

*I. Proof of Lemma 9*

Let $(A, B) \in \mathcal{Z}_\delta$ be fixed. Let us say node $i$ is *bad* if $|\mathsf{L}(i; A, B) - \mathsf{L}(i; A_R, B_R)| > \tau \log n$. Note that there are at most $\binom{n}{\delta n} \cdot 2^{\delta n}$ many partitions in $\mathcal{Z}_\delta$. As $\binom{n}{\delta n} \cdot 2^{\delta n} \le n^{\delta n} \cdot 2^{\delta n} \le e^{2\delta n \log n}$, it suffices to prove

$$\Pr\left( \sum_{i=1}^n \mathbb{I}\{i \text{ is bad}\} > \frac{\delta}{2} n \right) \le O\left( e^{-3 \cdot \delta n \log n} \right)$$

.

Note that standard concentration inequalities for indicator variables (e.g. Chernoff bound) cannot be directly applied to show this since $\{\mathbb{I}\{i \text{ is bad}\}\}_{i=1}^n$ are not mutually independent. For instance, the events [1 is bad] and [2 is bad] both depend on the occurrence of the edge $\{1, 2\}$. Inspired by [3], we resolve this issue via *decoupling* analysis. Observe the following upper bound

$$|\mathsf{L}(i; A, B) - \mathsf{L}(i; A_R, B_R)| = \log\left( \frac{(1-\beta)\alpha}{(1-\alpha)\beta} \right) |e(i, A) - e(i, A_R) - e(i, B) + e(i, B_R)|$$

$$= \log\left( \frac{(1-\beta)\alpha}{(1-\alpha)\beta} \right) |e(i, A \setminus A_R) - e(i, A_R \setminus A) - e(i, B \setminus B_R) + e(i, B_R \setminus B)|$$

$$\le \log\left( \frac{(1-\beta)\alpha}{(1-\alpha)\beta} \right) \{e(i, A \triangle A_R) + e(i, B \triangle B_R)\} = 2 \log\left( \frac{(1-\beta)\alpha}{(1-\alpha)\beta} \right) e(i, A \triangle A_R). \tag{23}$$

The last term can be split into two last term can be split into $2 \log\left( \frac{(1-\beta)\alpha}{(1-\alpha)\beta} \right) \Delta_i^{(1)} + 2 \log\left( \frac{(1-\beta)\alpha}{(1-\alpha)\beta} \right) \Delta_i^{(2)}$, where $\Delta_i^{(1)} := e(i, (A \triangle A_R) \cap \{1, 2, \cdots, i\})$ and $\Delta_i^{(2)} := e(i, (A \triangle A_R) \setminus \{1, 2, \cdots, i\})$. With this splitting, note that the collection of random variables $\left\{ \Delta_i^{(x)} \right\}_{i=1}^n$ is mutually independent for each $x = 1, 2$. Now, by letting $\mathbb{I}_i^{(x)} := \mathbb{I}\left\{ \Delta_i^{(x)} > \frac{\tau}{4 \log\left( \frac{(1-\beta)\alpha}{(1-\alpha)\beta} \right)} \log n \right\}$ for $x = 1, 2$, we obtain

$$\Pr\left( \sum_{i=1}^n \mathbb{I}\{i \text{ is bad}\} > \frac{\delta}{2} n \right) \overset{(a)}{\le} \Pr\left( \left[ \sum_{i=1}^n \mathbb{I}_i^{(1)} > \frac{\delta}{4} n \right] \bigcup \left[ \sum_{i=1}^n \mathbb{I}_i^{(2)} > \frac{\delta}{4} n \right] \right)$$

$$\le \Pr\left( \sum_{i=1}^n \mathbb{I}_i^{(1)} > \frac{\delta}{4} n \right) + \Pr\left( \sum_{i=1}^n \mathbb{I}_i^{(2)} > \frac{\delta}{4} n \right),$$

where $(a)$ is due to the fact that when both $\sum_{i=1}^n \mathbb{I}_i^{(1)} \le \frac{\delta}{4} n$ and $\sum_{i=1}^n \mathbb{I}_i^{(2)} \le \frac{\delta}{4} n$ are true, there could be at most $2 \cdot \frac{\delta}{4} n$ many bad nodes. Hence, it is sufficient to prove $\Pr\left( \sum_{i=1}^n \mathbb{I}_i^{(1)} > \frac{\delta}{4} n \right) \le O\left( e^{-3 \cdot \delta n \log n} \right)$.

**Lemma 12.** *For any constant $\ell > 0$, there exists $\delta_0 > 0$ such that the following holds: whenever $(A, B) \in \mathcal{Z}_\delta$ for $\delta < \delta_0$,* $\Pr\left( e(i, A \triangle A_R) > \frac{\tau}{4 \log\left( \frac{(1-\beta)\alpha}{(1-\alpha)\beta} \right)} \log n \right) \le 2n^{-\ell}$.

*Proof.* See Appendix A-L. $\qquad\qquad\qquad\qquad\qquad\qquad\qquad\qquad\qquad\qquad\qquad\qquad\qquad\qquad\qquad\square$

Let $\ell > 0$ be a constant to be chosen later. By Lemma 12, for $1 \le i \le n$ and $\delta < \delta_0$,

$$\Pr\left( \mathbb{I}_i^{(1)} = 1 \right) = \Pr\left( \Delta_i^{(1)} > \frac{\tau}{4 \log\left( \frac{(1-\beta)\alpha}{(1-\alpha)\beta} \right)} \log n \right) \le \Pr\left( e(i, A \triangle A_R) > \frac{\tau}{4 \log\left( \frac{(1-\beta)\alpha}{(1-\alpha)\beta} \right)} \log n \right) \le 2n^{-\ell}.$$

By Chernoff-Hoeffding [4],

$$\Pr\left( \sum_{i=1}^n \mathbb{I}_i^{(1)} > \frac{\delta}{4} n \right) \le \exp\left( -n \mathsf{D}_{\mathsf{KL}}\left( \frac{\delta}{4} \,\bigg\|\, 2n^{-\ell} \right) \right) \le \exp\left( -3\delta n \log n \right),$$

where the last inequality follows by taking $\ell > 0$ sufficiently large as follows:

$$\mathsf{D}_{\mathsf{KL}}\left( \frac{\delta}{4} \,\bigg\|\, 2n^{-\ell} \right) = \frac{\delta}{4} \log\left( \frac{\frac{\delta}{4}}{2n^{-\ell}} \right) + \left( 1 - \frac{\delta}{4} \right) \log\left( \frac{1 - \frac{\delta}{4}}{1 - 2n^{-\ell}} \right)$$

$$\overset{(a)}{\ge} \frac{\delta}{4} \log\left( \frac{\frac{\delta}{4}}{2n^{-\ell}} \right) + \log\left( \frac{1 - \frac{\delta}{4}}{1 - 2n^{-\ell}} \right) \ge \frac{\delta}{4} \log\left( \frac{\frac{\delta}{4}}{2n^{-\ell}} \right) + \log\left( 1 - \frac{\delta}{4} \right)$$

$$\overset{(b)}{=} \frac{\delta}{4} \log\left( \frac{\delta n^\ell}{8} \right) - \frac{\delta}{4} + O\left( \delta^2 \right) \overset{(c)}{\ge} \frac{\delta}{4} \cdot \{(\ell - 1) \cdot \log n - (\ell - 1) \cdot \log 8 - 1 + O(\delta)\}$$

where $(a)$ follows from the fact that for $\ell > 1$, $2n^{-\ell} \leq n^{-1} \leq \frac{\delta}{4}$, implying $\log\left(\frac{1-\frac{\delta}{4}}{1-2n^{-\ell}}\right) < 0$; $(b)$ is due to the fact $\log(1-x) = -x + O(x^2)$ as $x \to 0$; $(c)$ follows from the fact that $\delta \geq n^{-1}$. The last term can be surely made greater than $3\delta \log n$ by taking $\ell > 0$ sufficiently large.

### J. Proof of Lemma 10

By definition, $\left|\widehat{\mathsf{L}}(i; A, B) - \mathsf{L}(i; A, B)\right|$ is bounded by

$$|c_s - \hat{c}_s| \cdot \{e(\{i\}, A) + e(\{i\}, B)\} + |c_r - \hat{c}_r| \cdot \{\Pi_i(u_R) + \Pi_i(v_R)\} . \tag{24}$$

First, note that for any subset $A$, $e(\{i\}, A)$ is bounded by $\deg(i) := e(\{i\}, [n] \setminus \{i\})$. Moreover, $\Pi_i(u_R)$ and $\Pi_i(u_R)$ are both bounded by $\Omega_i := \{j : (i, j) \in \Omega\}$. Since we are in the regime where $n\alpha = \Theta(\log n)$, one can use a standard large deviation inequality to prove that there exist a constant $c_1 > 0$ such that with high probability, $\deg(i) \leq c_1 \log n$ for any $i$. More specifically, $\deg(i)$ is statistically dominated by $\sum_{i=1}^{n-1} A_i$, where $A_i \overset{\text{i.i.d.}}{\sim} \text{Bern}(\alpha)$. Hence, $\Pr\left(\deg(i) > t\right) \leq \Pr\left(\sum_{i=1}^{n-1} A_i > t\right)$, and by Bernstein inequality, the last term is bounded by $2\exp\left(\frac{-\frac{1}{2}t^2}{(n-1)\alpha+t}\right)$. By choosing $t = c_1 \log n$ for sufficiently large $c_1 > 0$, one can ensure $\Pr\left(\deg(i) > c_1 \log n\right) = o(n^{-1})$, which finish shows the claim after taking the union bound. Similarly, since $p = \Theta\left(\frac{\log n}{m} + \frac{\log m}{n}\right)$ and $m = O(n)$, $mp = \Theta(\log n)$, and hence, one can show that there exists $c_2 > 0$ such that $\Omega_i \leq c_2 \log n$ for any $i$.

Hence, (24) is bounded by $2c_1 \log n |c_s - \hat{c}_s| + 2c_2 \log n |c_r - \hat{c}_r|$, meaning it is sufficient to show (i) $|c_s - \hat{c}_s| \leq \frac{\tau}{4c_1}$ and (ii) $|c_r - \hat{c}_r| \leq \frac{\tau}{4c_2}$. The proof of (ii) is similar to (i), and indeed easier, so we omit the proof. Note that $|c_s - \hat{c}_s| \leq \left|\log\frac{\hat{\alpha}}{\alpha}\right| + \left|\log\frac{\hat{\beta}}{\beta}\right| + \left|\log\frac{1-\hat{\alpha}}{1-\alpha}\right| + \left|\log\frac{1-\hat{\beta}}{1-\beta}\right|$, which is equal to $\left|\log\left(1 + \frac{\hat{\alpha}-\alpha}{\alpha}\right)\right| + \left|\log\left(1 + \frac{\hat{\beta}-\beta}{\beta}\right)\right| + \left|\log\left(1 - \frac{\hat{\alpha}-\alpha}{1-\alpha}\right)\right| + \left|\log\left(1 - \frac{\hat{\beta}-\beta}{1-\beta}\right)\right|$.

**Lemma 13.** *Let $\eta := \frac{|A_1^{(0)} \setminus A_R|}{n}$. If $\eta$ is sufficiently small, we have $\left|\frac{\hat{\alpha}-\alpha}{\alpha}\right| = O(\eta)$ and $\left|\frac{\hat{\beta}-\beta}{\beta}\right| = O(\eta)$ with probability $1 - o(1)$.*

*Proof.* See Appendix A-M. $\square$

Due to Lemma 13, the last term is $O(\eta)$, and as the analysis of Stage 1 guarantees that $\eta = o(1)$, the proof is completed.

### K. Proof of Lemma 11

From (19) and (20), it follows that $\mathbb{M}_{X_1}(1/2) = \left(\sqrt{\alpha\beta} + \sqrt{(1-\alpha)(1-\beta)}\right)^2$ and $\mathbb{M}_{Y_1}(1/2) = 2p\sqrt{\theta(1-\theta)} + 1 - p$.

Simple computations yield the distribution of $V_1$ and $W_1$: For simplicity, let $c_g := \log\left(\frac{(1-\beta)\alpha}{(1-\alpha)\beta}\right)$ and $c_m := \log\left(\frac{1-\theta}{\theta}\right)$. Then, we have: $\Pr\left(V_1 = c_g\right) = \Pr\left(V_1 = -c_g\right) = \frac{\sqrt{(1-\alpha)(1-\beta)\alpha\beta}}{\left(\sqrt{\alpha\beta}+\sqrt{(1-\alpha)(1-\beta)}\right)^2}$ and $\Pr\left(V_1 = 0\right) = \frac{\alpha\beta+(1-\alpha)(1-\beta)}{\left(\sqrt{\alpha\beta}+\sqrt{(1-\alpha)(1-\beta)}\right)^2}$; $\Pr\left(W_1 = c_m\right) = \Pr\left(W_1 = -c_m\right) = \frac{p\sqrt{\theta(1-\theta)}}{2p\sqrt{\theta(1-\theta)}+1-p}$ and $\Pr\left(W_1 = 0\right) = \frac{1-p}{2p\sqrt{\theta(1-\theta)}+1-p}$.

We now compute the second moments of $V_1$ and $W_1$: $\mathbb{E}[V_1^2] = c_g^2 \frac{\sqrt{(1-\alpha)(1-\beta)\alpha\beta}}{\left(\sqrt{\alpha\beta}+\sqrt{(1-\alpha)(1-\beta)}\right)^2} = O(\sqrt{\alpha\beta})$ and $\mathbb{E}[W_1^2] = c_m^2 \frac{p\sqrt{\theta(1-\theta)}}{2p\sqrt{\theta(1-\theta)}+1-p} = O(p)$. Hence, by Chebyshev's inequality (note that $V_i$'s and $W_j$'s are mean zeros),

$$\Pr\left(\sum_{i=1}^{K} V_i + \sum_{j=1}^{L} W_j > (pL)^{3/4}\right) \leq \frac{\sum_{i=1}^{K}\mathbb{E}[V_i^2] + \sum_{j=1}^{L}\mathbb{E}[W_j^2]}{(pL)^{3/2}}$$

$$= \frac{O(\sqrt{\alpha\beta}K) + O(pL)}{(pL)^{3/2}} \overset{(a)}{\leq} \frac{O(pL)}{(pL)^{3/2}} \overset{(b)}{\to} 0,$$

where $(a)$ follows from the fact that $pL \geq \sqrt{\alpha\beta}K$; $(b)$ follows from the fact that $pL = \omega(1)$.

### L. Proof of Lemma 12

As $(A, B) \in \mathcal{Z}_\delta$, $e(i, A \triangle A_R)$ is a sum of at most $\delta n$ independent random variables each of which has distribution either $\text{Bern}(\alpha)$ or $\text{Bern}(\beta)$. Hence, $e(i, A \triangle A_R)$ is statistically dominated by $\sum_{i=1}^{\delta n} A_i$, where $A_i \overset{\text{i.i.d.}}{\sim} \text{Bern}(\alpha)$. Thus,

$$\Pr\left(e(i, A \triangle A_R) > \frac{\tau}{4\log\left(\frac{(1-\beta)\alpha}{(1-\alpha)\beta}\right)} \log n\right) \leq \Pr\left(\sum_{i=1}^{\delta n} A_i > \frac{\tau}{4\log\left(\frac{(1-\beta)\alpha}{(1-\alpha)\beta}\right)} \log n\right) .$$

On the other hand, by Lemma 7 with $k = \max\left\{5e, \ \ell \cdot \frac{2\log n}{n\alpha}\right\}$[4], one obtains

$$\Pr\left(\sum_{i=1}^{\delta n} A_i \geq \frac{kn\alpha}{\log\frac{1}{\delta}}\right) \leq 2\exp\left(-\frac{kn\alpha}{2}\right) \leq 2n^{-\ell}.$$

As $\lim_{\delta\to 0+}\frac{1}{\log\frac{1}{\delta}} = 0$, by taking $\delta_0$ sufficiently small, one can ensure $\frac{kn\alpha}{\log\frac{1}{\delta}} \leq \frac{\tau}{4\log\left(\frac{(1-\beta)\alpha}{(1-\alpha)\beta}\right)}\log n$ whenever $\delta < \delta_0$, which completes the proof.

*M. Proof of Lemma 13*

We will only prove $\left|\frac{\hat{\alpha}-\alpha}{\alpha}\right| = O(\eta)$; the proof of $\left|\frac{\hat{\beta}-\beta}{\beta}\right| = O(\eta)$ follows similarly. Note that $e(A_1^{(0)}, A_1^{(0)}) = \sum_{i=1}^{\binom{(1-\eta)n}{2}+\binom{\eta n}{2}} A_i + \sum_{i=1}^{\binom{n}{2}-\binom{(1-\eta)n}{2}-\binom{\eta n}{2}} B_i$, where $A_i \overset{\text{i.i.d.}}{\sim} \text{Bern}(\alpha)$ and $B_i \overset{\text{i.i.d.}}{\sim} \text{Bern}(\beta)$. Let $\gamma := \frac{\binom{n}{2}-\binom{(1-\eta)n}{2}-\binom{\eta n}{2}}{\binom{n}{2}}$. Then, it is straightforward to check that $\gamma = O(\eta)$. By triangle inequality, $\left|\frac{1}{2}\alpha - \frac{e(A_1^{(0)}, A_1^{(0)})}{2\binom{n/2}{2}}\right| \leq \left|\frac{(1-\gamma)}{2}\alpha - \frac{1}{2\binom{n}{2}}\sum_{i=1}^{(1-\gamma)\binom{n}{2}} A_i\right| + \frac{\gamma}{2}\alpha + \frac{1}{2\binom{n}{2}}\sum_{i=1}^{\gamma\binom{n}{2}} B_i$.

Simple applications of Bernstein inequality deduce that with high probability, $\left|\frac{(1-\gamma)}{2}\alpha - \frac{1}{2\binom{n}{2}}\sum_{i=1}^{(1-\gamma)\binom{n}{2}} A_i\right| = O\left(\frac{\log n}{n^{3/2}}\right)$ and $\frac{1}{\gamma\binom{n}{2}}\sum_{i=1}^{\gamma\binom{n}{2}} B_i \leq 2\beta = O(\alpha)$. Taking collectively, we have $\left|\frac{1}{2}\alpha - \frac{e(A_1^{(0)}, A_1^{(0)})}{2\binom{n/2}{2}}\right| \leq O(\gamma)\alpha$. Similarly, one can show $\left|\frac{1}{2}\alpha - \frac{e(A_2^{(0)}, A_2^{(0)})}{2\binom{n/2}{2}}\right| \leq O(\gamma)\alpha$, which concludes the proof.