[Reviews · NeurIPS 2018]

Reviewer 1



Summary of paper: The paper considers the problem of completing a binary rating matrix, and theoretically studies whether side information in terms of graphs among users or items aid in the matrix completion task. On a technical level, the study assumes that the users belong to two communities, and users in the same group rate all items identically (upto some noise). The community structure is further revealed by the side information in terms of a graph generated from stochastic block model. The paper derives the optimal sample complexity for rating estimation in terms of the expected number of ratings that needs to be seen to exactly recover the rating matrix. Review summary: The paper is a solid piece of theoretical work and clearly written. The problem tackled is quite important, and the paper clearly shows how much graph side information helps under the assumed setting. It is known that side information helps but a clear quantification of its benefits is not there in existing literature. On the negative side, it seems that the model and the method are too restrictive for the general rating estimation problem. Compared to general methods like Rao et al [42], this work seems too fine-tuned for the theoretical problem (with clustered users) and has limited direct impact on practice. Originality: The theory and algorithm relies heavily on the recent literature on block models. The use of this theory in rating estimation is somewhat novel, although some recent works have made similar connections in related problems (for example, [R1]). Significance and quality: The main positive for the paper is its significance. Quantifying the usefulness of side information in any learning problem is a difficult task. The paper achieves this goal by presenting and analysing a formal problem, which shows that the optimal sample complexity is indeed lower if side information is used. The theory looks sound, but I have not read the proofs in the supplementary. Apart from the correctness, I have a fundamental question: - it seems that graph side information only aids when the quantity I_s is of the order log(n)/n. Doesn’t this imply that side information does not provide significant advantages? That is, we cannot drastically reduce the sample complexity by using side graphs (unless parameters are fine-tuned as in Example 2). The practicality of the work is rather limited. The model assumes that users are clustered into two groups, the graph shows exactly these two groups, and the spectral algorithm mainly tries to find the groups. Hence, in comparison to general methods like Rao et al [42], this work seems too fine-tuned to the theoretical problem (with clustered users). - can methods like [42] that simply use the graph as a regulariser achieve near optimality in this setting? I like the fact that the proposed spectral method adaptively estimates some parameters, though I am not sure how c1, c2 can be estimated. However, as noted above, the methods loses generality as it mainly uses the graph for its cluster structure and not as some sort of regulariser (side graphs without community structure don't contribute). The numerical section is bit disappointing, which is fine for a theory paper. The networks are real, but the ratings are synthetically generated based on the ground truth communities. - can we use the proposed method for any real binary estimation problem? - the numerical comparisons are perhaps bit unfair since the study is on a setup where finding the cluster structure from graph is supposed to reveal correct ratings, and hence, proposed method outperforms Figure 2 is supposed to show that there is a clear phase transition (white to black) at the derived theoretical sample complexity (orange line). Given how convincing empirical phase transition plots can be for some problems, I am not sure if Figure 2 validates the sudden transition proved in Theorem 1. Please clarify. Clarity: The paper is generally well-written. Since this is a theory paper in a general ML conference, it would be great if the authors can add some discussions on the practical consequences of the results. For example: - I_s of the order log(n)/n is only regime when side information helps. Why doesn’t it help to have more dense disjoint graphs? This is practically counter-intuitive. - when does it not matter to have side information? Why? - what is the implication of exact recovery in practice? Minor comments: - line 67: “out-of-reach” is too strong, perhaps “not studied” - line 205: “cluster size” should be “number of clusters" - line 220: should w be \omega? - line 223: footnote mark 3 looks confusing after the (log n). [R1] Lee and Shah. Reducing crowdsourcing to graphon estimation, statistically. AISTATS 2018 UPDATE AFTER AUTHOR REBUTTAL: I thank the authors for their response. It will be great if they can add the mentioned discussions to the paper. On a general note, I would also request the authors to include some concluding remarks about the limitation of the study to clustered user scenario, and if (or how) one can extend these results to more general graph side information.

Reviewer 2



Motivation: My main concern is the motivation of the method. To recover the missing rates in the binary matrix, the authors propose several strict assumptions, which call for reconsideration on whether it is practical: only two clusters exist with the same size and users in the same cluster share the same ratings over items. Finally, the partially observed matrix could be a noisy rank-1 matrix. Formulation: --> I am confused about the connections between the worst-case probability of error and the optimal sample complexity. If lim_{n->infty}inf max Pr(phi(N,G\)neq R’)=0, it is still questionable to conclude that the recovery can be guaranteed. It might needs more analytical investigation but I suggest the authors consider defining the probability of error as max_{R’:||u_{R’}-v_{R’}||_0 \leq \gamma m } Pr(\phi(N,G)\neq R’). --> In Theorem 1, It seems unclear on the parameter p . How do the assumptions work in Th1? In terms of the main contribution of the paper, it would be better to discuss more on the theoretical bound in the Th1, especially on every parameter associated with the bound. Experiment: More related side information matrix recovery methods should be compared. Writing: 1. What do u_R and v_R exactly mean? 2. On page 3, line 130, what does the nmp^* mean? -------------------------------------------------------------- Update: This paper has several strong points, especially on the theoretical recovery sampling rate with graph side information. However, as a real-world problem-driven method for the recommendation system, but not a pure theoretical toy-problem, I feel not that confident to fully recommend this paper. First, there are many existing matrix completion with side info methods that need discussion in depth in this paper. E.g., the paper named "Chiang, Kai-Yang, Cho-Jui Hsieh, and Inderjit S. Dhillon. "Matrix completion with noisy side information." Advances in Neural Information Processing Systems. 2015" has a more relaxed assumption on the target matrix and their optimal and worst theoretical sampling rates are given in their paper. Plus, to be more fair comparison, the experiment is suggested to include more matrix completion with side info methods, at least several regularization based methods cited by the authors. I appreciate the efforts on the theoretical aspect in this paper, but in my view, the proposed theory seems still far away from the nature of a practical recommendation system. Rather than the theory with a strong assumption, I would be more convinced to see the comprehensive experiments showing the effectiveness and efficiency of the method first. Overall, I would recommend this paper if the paper targets more at discussing a pure math toy-problem. But my rating is still marginally below the acceptance threshold.

Reviewer 3



This paper introduces a problem where both a stochastic block model graph and a partial ratings matrix (with the same community structure) are observed, and the goal is to fill in the missing ratings. Sharp information-theoretic thresholds are derived, and a practical algorithm is shown to suffice for the upper bound. Overall, my impression of the paper is positive. The main limitation is that the model is very simple (e.g., two equal-size communities, identical ratings within a community). However, such a model is still interesting from a theoretical viewpoint, and I view it as being analogous to the role of the (also very simple) stochastic block model that sparked a lot of interesting research for community detection. Basically, I think it is quite suitable for an initial primarily-theoretical work to adopt such a simple model, whose assumptions can potentially be relaxed in future works. Because of the length of the supplementary material, I am unable to be confident about the correctness of all results. But I found myself fairly convinced regarding the information-theoretic limits, and I did not notice any obvious problems in the analysis of the practical algorithm. The writing is quite dense. It feels like a lot is packed into the main body, with lots of long inline equations and so on. This is not a major problem, but it makes it a bit harder to read. A technical comment: The first chain of display equations of the full proof of the converse (bottom of supplementary p4) needs revising. Most importantly, equations (a), (b), and (c) need a \sum_{R'} Pr[R'] out front. (I guess this was the intended meaning anyway, but the notation is not good) Despite this, I am confident of the claim itself on the ML estimator. Here are a number of more specific comments (all fairly minor): - First of the abstract sounds a bit vague, and is perhaps not the best start - p1: "Since then" -- it's unclear what this is referring to - p2: The comparison to [42] makes it sound like your models are the same, which I guess they might not be. - p2: Shortly after "Observation model.", remove "the" - p2: after "exact recovery is the most difficult", perhaps there should be a "subject to...". Letting all entries be 0 would be most difficult! - Theorem 1: Perhaps factor (1+epsilon) and (1-epsilon) outside the max{.} - Sometimes confusion between "notion" and "notation" - p4: Perhaps avoid notation like (1)= and (2)= - p5: Please cite the power method runtime - p6: I definitely suggest explicitly defining the ML estimator at this early stage, since at this stage its definition is not completely clear (e.g., do you limit to matrices consistent with the gamma*n constraint?) - Footnote 2: Perhaps expand this sentence to say that the two sets of assumptions still overlap considerably. Otherwise it might sound like the two theorems are not comparable. - p6: Please state the the full proofs are in the supplementary material - p6: Perhaps instead of just "WLOG" mention symmetry? And again on p7. - p7: "for the time being" sounds strange - p7: Why are there |.| around S(z) ? - p8: Typos "showing [the] practicality" and "state of the arts" (the latter is also in the abstract) - Supplementary p2: Typo "We will proof" - Supplementary Theorem 1: The converse should say for ANY \psi (not "for some") - Supplementary p7: It is misleading to say that |F| = \eta * n. The = should be <= (e.g., maybe we get lucky and have perfect recovery) - Supplementary p9: The hats are misplaced on c_s and c_r ===== POST-REBUTTAL: I am satisfied with the author responses. I still suggest using more explicit notation for averaging around the R' part, like \sum_{r'} Pr[R' = r']